# Effect of physical activity on the risk of frailty: A systematic review and meta-analysis

**Wenjing Zhao[1,2☯], Peng Hu[3,4☯], Weidi Sun[5], Weidong Wu[6], Jinhua Zhang[7], Hai Deng[8], Jun Huang[9], Shigekazu Ukawa[10], Jiahai Lu[3], Akiko Tamakoshi[2]\*, Xudong Liu[4]\***

**1** School of Public Health and Emergency Management, Southern University of Science and Technology, Shenzhen, China, **2** Department of Public Health, Faculty of Medicine, Hokkaido University, Sapporo, Japan, **3** Department of Epidemiology, School of Public Health, Sun Yat-sen University, Guangzhou, China, **4** School of Public Health, Guangdong Pharmaceutical University, Guangzhou, China, **5** School of Public Health and Women's Hospital, Zhejiang University School of Medicine, Hangzhou, China, **6** School of Public Health, Xinxiang Medical University, Xinxiang, China, **7** School of Nursing, Xinxiang Medical University, Xinxiang, China, **8** Department of Cardiology, Guangdong Cardiovascular Institute, Guangdong Provincial People's Hospital, Guangdong Academy of Medical Science, Guangzhou, China, **9** Department of Geriatrics, Institute of Geriatrics, Guangdong Provincial People's Hospital, Guangdong Academy of Medical Science, Guangzhou, China, **10** Research Unit of Advanced Interdisciplinary Care Science, Graduate School of Human Life Science, Osaka City University, Osaka, Japan

☯ These authors contributed equally to this work.
\* xdliu.cn@hotmail.com (XL); tamaa@med.hokudai.ac.jp (AT)

**Data Availability Statement:** This study is a systematic review and meta-analysis; the data was extracted from published research. The data is

## Abstract

### Objective

The relationship between physical activity (PA) and the risk of frailty has not reached a conclusive result. This systematic review with meta-analysis aimed to evaluate the effect of PA on the onset of frailty in the community-dwelling middle and older age adults by pooling data from cohort studies.

### Methods

A systematic literature search was performed via PubMed, Embase, and Web of Science up to June 01, 2021. Pooled adjusted effect estimates (ES) with 95% confidence interval (CI) were calculated by using the random-effect model and by comparing the highest with lowest levels of PA. Heterogeneity was tested using the $I^2$ statistic and Q-test. The quality of evidence was evaluated by using the Grading of Recommendations Assessment, Development and Evaluation (GRADE) approach.

### Results

A total of ten cohort studies with 14 records were selected, and the GRADE approach classified the quality of evidence as low. In comparison with the lowest level of PA, the highest level of PA was associated with 41% decreased odds of frailty (ES: 0.59, 95% CI: 0.51–0.67; $I^2$ = 70.0%, $P_{\text{-heterogeneity}}$ < 0.001) after pooling results from included studies. In stratified analysis by frailty assessment approach, the highest level of PA was significantly associated with 37% (ES 0.63, 95% CI: 0.52–0.77, 49% (ES: 0.51, 95% CI: 0.41–0.63), and 30% (ES: 0.70, 95% CI: 0.65–0.75) reduced odds of frailty when pooling studies using criteria of

available by contacting the corresponding author or extracting from original published research.

**Funding:** This study was supported by the Science and Technology Program of Guangzhou City (No.202102080404), the Guangdong Basic and Applied Basic Research Foundation (No. 2022A1515010686), and the Univers Foundation (No.17-02-160). The funders had no roles in the design, analysis, or writing of this manuscript. There was no other additional external funding received for this study.

**Competing interests:** The authors have declared that no competing interests exist.

physical frailty, multidimensional model, and accumulation of disability, respectively. Stratified analyses further by PA indicators and PA assessment tools yielded similar protective effects in any subgroups.

## Conclusions

This study with moderate-certainty evidence shows that a higher level of PA was associated with lower odds of frailty, and the benefits of PA for frailty prevention were independent of frailty assessment tools, PA indicators, and PA assessment methods. Findings from this study may help implement active exercise strategies to prevent frailty.

## Introduction

Frailty is a transition state from healthy aging to disability. Age-related cumulative declines in physiological reserve and vulnerability to stressors promote the presence of frailty [1]. It is estimated that more than 10% of adults aged 65 years or above suffered from frailty in the world [2, 3] and the prevalence was even higher in the individuals living with cancer, heart failure, and other chronic diseases [4–6]. The frail adults aged 60 years or older living in the community are increasing globally at a speed of around 43 new cases per 1000 person-years [7]. Frailty is closely related to a range of adverse health-related events of falling [8], fractures [9], disability [10], hospitalization [11], and even mortality [12] in older residents. The failure to prevent frailty will continue to aggravate the burden of health care costs for individuals and countries [13, 14].

Physical activity (PA) can regulate the function of multiple systems in the body [1]; the dysfunction of these body system can further contribute to the development of frailty [1]. A systematic review [15], a scoping review [16], and a systematic review and meta-analysis [17] synthesized the evidence and found that exercise training can reduce the level of frailty and improve the prognosis of frailty among older adults. However, epidemiological studies on the relationship between PA and frailty did not reach to a consistent conclusion. For instance, Trombetti [18] and Pérez-Tasigchana [19] did not find any association between PA and frailty, while Borda [20] and Savela [21] reported that PA was a protective factor for frailty. A meta-analysis found that physical exercise therapy could improve mobility and physical functioning in elderly patients suffering from mobility problems, disability and/or multi-morbidity [22]; however, more original studies have been reported in the following decade, making it necessary to update the synthetic evidence. A more recent systematic review including four randomized controlled trials and two prospective cohort studies found that physical activity might be an effective intervention for preventing frailty mong people aged 65 years and older [23]; nevertheless, the intervention measures varied largely among four RCTs and one included RCT adopted combined nutrition and exercise interventions, limiting the application of the results. Moreover, this study [23] only included two cohort studies, but more cohort studies have been reported [18–21, 24–30]. In addition, there are many evaluation methods for frailty, and no unified standard has been formed [31]. Whether different methods of frailty assessment influence the association between PA and frailty onset is unclear.

Pan and colleagues published a registered protocol for systematic review and meta-analysis to demonstrate the association between PA and the risk of frailty in the old community-dwelling residents [32]; however, the related systematic review and meta-analysis has not

been reported in the past three years. What's more, this protocol [32] did not mention how to define frailty and whether synthesize the evidence by frailty definition or not.

Therefore, we conducted this systematic review and meta-analysis by synthesize evidence from cohort studies to examine the effect of PA on the onset of frailty among the community-dwelling middle and older age adults, so to provide evidence for frailty prevention for middle and order age adults.

## Methods

### Search strategy and selection criteria

This study was implemented according to the PRISMA guideline [33]. Three researchers (PH, WJZ, WDS) searched the literature independently. The group discussion with the other two researchers (XDL, TA) was conducted to resolve the dissidence. The literature was systematically searched from the inception to June 01, 2021, through three electronic databases of PubMed, Embase, and Web of science. The keywords and retrieval strategy were as follows: ('physical activity' OR 'exercise' OR 'acute exercise' OR 'isometric exercises' OR 'aerobic exercise' OR 'physical exercise' OR 'endurance exercise' OR 'resistance exercise' OR 'strength exercises' OR 'training' OR 'exercise training', OR 'combined training', OR 'weight-lifting' OR 'running' OR 'jogging' OR 'swimming' OR 'walking' OR 'yoga' OR 'Tai chi' OR 'daily activity' OR 'lifestyle' OR 'sport') AND ('frailty' OR 'frailness' OR 'frailty syndrome' OR and 'debility'). A reverse reference citation tracking was also carried out to search for the possible literature. More details of the search strategy were shown in S1 Table.

Inclusion criteria were as follows: the exposure of PA including exposure intensity, frequency, duration, volume, step, or any specific type (such as leisure-time physical activities, occupational activities, and exercise, etc.) was reported; frailty and its assessment approaches (physical frailty, multidimensional approach, accumulation of disability, etc.) were reported; design of cohort studies; middle and older age healthy adults; the relationship between PA and frailty was evaluated; the study was published in English. Cross-sectional studies, animal studies, trials, reviews, editorials, letters, abstracts, and studies lacking data to manifest the relationship between PA and frailty were excluded after reviewing title, abstract or full-text.

### Data extraction and quality assessment

The detailed information was extracted from the eligible studies including the first author of the studies, the publication year, study design, region, sample size, age (mean or median, range), the proportion of female individuals, the follow-up period, confounders, frailty assessment approach, PA assessment method, PA indicator, comparison of different PA levels and the corresponding effect estimates.

The quality of the included studies was assessed by using criteria of the Newcastle-Ottawa Scale (NOS) [34]. The maximum total score for NOS was 9 stars, including 4 for selection, 2 for comparability, and 3 for exposures assessment. A study would be classified as the quality of high (7–9 stars), moderate (4–6 stars), or low (1–3 stars) according to the total score obtained. The Grading of Recommendations Assessment, Development and Evaluation (GRADE) approach was used to evaluated the quality of the body of evidence [35]. The GRADE approach was based on considerations such as study design, risk of bias, inconsistency, imprecision, indirectness, publication bias and other aspects reported by the included studies. The quality of the evidence was characterized as high, moderate, low, or very low.

### Ethics statement

This study was approved from the Ethical Review Committee for Biomedical Research, School of Public Health, Sun Yat-sen University and from the Ethics Review Committee of Hokkaido University Graduate School of Medicine. The study was performed in accordance with the Declaration of Helsinki. The study was a systematic review and meta-analysis, the consent was waived for the study, and no patients were involved.

### Statistical analysis

A meta-analysis was conducted to estimate the overall pooled effect estimates (ES) based on the adjusted odds ratio or hazard ratio and 95% confidence interval (CI), by comparing the highest with the lowest levels of PA [36]. For those studies which reported results from the comparison of the lowest with highest categories, the reciprocal method was used to transform the original effect [37]. The heterogeneity was assessed by using $I^2$ index and Q-test. $I^2$ value of more than 50% or $P$-value of Q-test less than 0.05 indicated heterogeneity, and then the random-effect model was used to calculate pooled ES; otherwise, the fixed-effect model would be applied [38]. Repeated analysis was further conducted to evaluate the contribution of each study to the pooled effect by omitting one study at a time. To seek the potential factors influencing the association of frailty risk with PA, subgroup analysis and meta-regression analysis were conducted based on the characteristics of the region (Europe and America, Asia), sample size ($<$ 1000, $\geq$ 1000), female proportion ($<$ 50%, $\geq$ 50%), age ($<$ 70 years, $\geq$ 70 years), frailty assessment approach (physical frailty, accumulation of disability, or multidimensional approach), follow-up period ($<$ 10 years, $\geq$ 10 years), PA indicators (Volume, frequency, intensity, duration, steps), PA assessment methods (Questionnaire or Uniaxial accelerometry sensor), and effect estimates (odds ratio, hazard ratio). The publication bias was analyzed by the Begg's test, the Egger's test, and the funnel plot. The $P$-value of the Begg's test and the Egger's test lower than 0.05 or the asymmetric funnel plot suggested the publication bias. All statistical analyses were conducted by using Stata 12.0 software (Stata Corporation, College Station, TX, USA).

## Result

The study selection procedure is shown in Fig 1. The systematic review identified 8,193 articles from three electronic databases. About 1,749 articles were removed for duplication and 6,444 articles were removed after screening the title and the abstract. Among the 20 articles for further full-text reviewing, eight articles were excluded because the effect estimates were not provided or cannot be calculated, two were excluded for the cross-sectional design. Finally, 10 cohort articles [19–21, 24–30] were included in this systematic review and meta-analysis. The quality score of each study ranged from 6 to 9, depicting a moderate to a higher quality of included studies (S2 Table). By using GRADE approach, it is found that substantial heterogeneity was the main reason responsible for the limited quality of the evidence, whereas all plausible confounding factors were considered (S3 Table). Hence, the quality of the evidence from the outcomes evaluated by the GRADE system was assessed as low as a whole.

The general characteristics of the included studies are illustrated in Table 1. Yu et al. [30] reported the studies from three prospective cohorts in Hong Kong, Taiwan-urban and Taiwan-rural, so we divided this study into three cohorts for meta-analysis; Yuki et al. [28] evaluated the relationship between three dimensions of physical activity (daily number of walking steps, time of light-intensity physical activity, and time of moderate-to-vigorous intensity physical activity) and frailty risk among Japanese, and therefore we divided it into three records when doing meta-analysis. Among the other eight included studies, two studies were

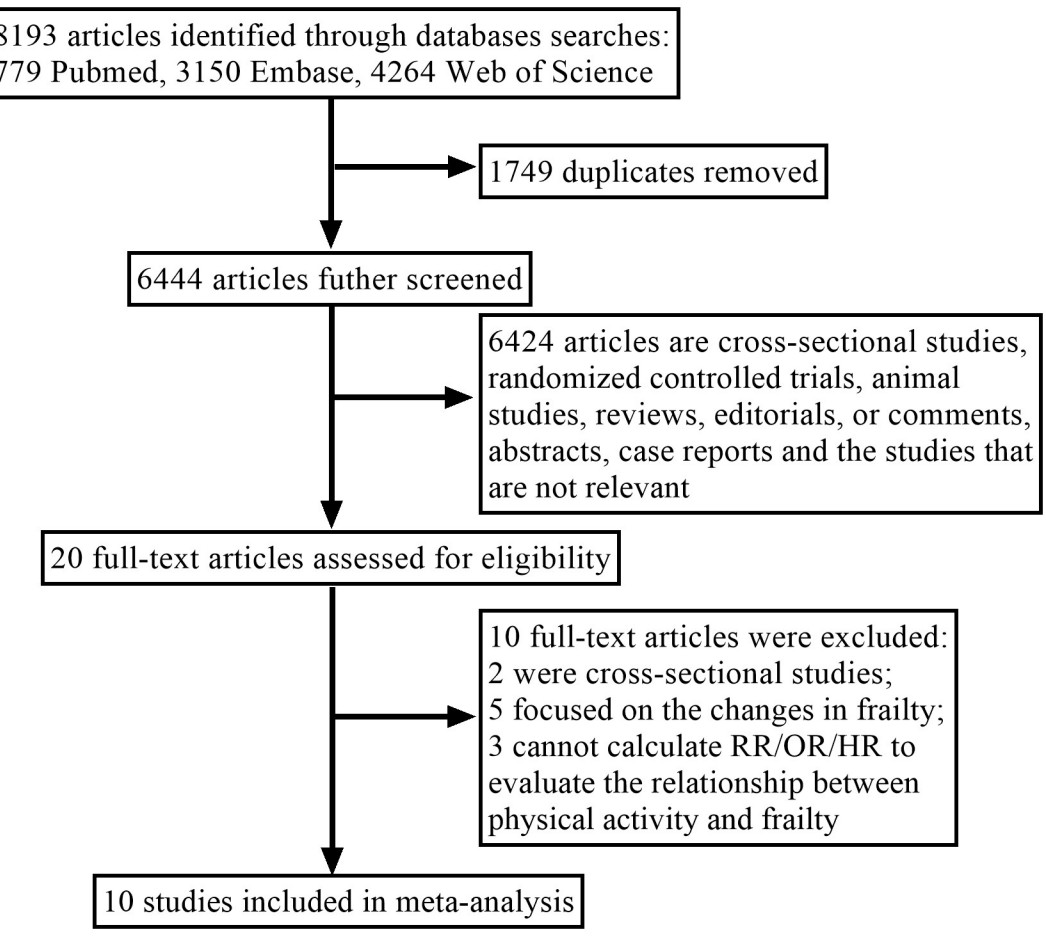

**Fig 1. Flow chart of the study selection process.**

from Finland [21, 27], two from the UK [26, 29], one from Japan [25], one from the USA [24], one from Mexico [20], and one from Spain [19]. A total of 34,943 participants were included in this meta-analysis. The total samples of each study ranged from 401 to 7,420, and the frailty cases ranged from 48 to 2,300. The mean age of the participants in each study ranged from 47.5 to 75.38 years. One study only included male participants [21], and the other nine studies included both genders (approximately 44.40%–59.2% were females) [19, 20, 24–30]. The median follow-up year ranged from 3 years to 26 years. Two studies used hazard ratio to display the effect [26, 29] and others used odds ratio.

Frailty was evaluated by Fried's or modified Fried's criteria in four studies [21, 27–29], by Gill frailty model in one study [24], by questionnaires of the Kaigo-Yobo Checklist in one study [25], by the FRAIL scale in one study [19], by the multidimensional approach in two studies [26, 30], and by the accumulation of disability in one study [20]. The scales of the Kaigo-Yobo Checklist and the FRAIL scale were validated by Fried's criteria and Gill frailty model was defined by the functional limitations; therefore, frailty assessed by Fried's criteria, modified Fried's criteria and Gill frailty model was considered as physical frailty [19, 21, 24, 25, 27–29]. One study with three records assessed PA with a uniaxial accelerometry sensor [28], and the other nine studies assessed PA with a questionnaire [19–21, 24–27, 29, 30]. Of the nine studies [19–21, 24–27, 29, 30], two studies evaluated PA with the common and official

**Table 1. General characteristics of the included studies\*.**

| Author, year, region | Total samples/, n | Frailty cases, n | Age, years, mean (SD), range | Female, n (%) | Follow-up, years, mean (SD) | Frailty assessment approach* | Frailty assessment approach classification | Statistical analysis | PA assessment method, PA indicator [comparison], evaluation year* | Effect estimator | Transformation of effect estimator by using the reciprocal of the original value† | Confounders |
|---|---|---|---|---|---|---|---|---|---|---|---|---|
| Pérez-Tasigchana et al. [19] 2020, Spain, Cohort Study | 1038 | 107 | 70, ≥60 | 59.2 | 8 | The FRAIL scale (fatigue, resistance, ambulation, weight loss, comorbidity) | Physical frailty | Logistic regression | Questionnaire: PA volume [vigorous/moderate Yes vs. Not], (2001) | OR 0.59 (0.34–1.01) | - | Age, gender, educational level, occupational status, BMI, abdominal obesity, hypertension, hypercholesterolemia, coronary heart disease, stroke, diabetes, history of hip fracture, cancer, and osteoarthritis |
| Borda et al. [20] 2020 Mexico Cohort Study | 6087 | 2300 | 62.2 (8.5), >60 | (44.8) | 3 | Frailty index (FI), (self-rated health, current health compared with prior health status, functional assessments self-reported chronic diseases, difficulty in basic ADL and instrumental ADL, self-reported common symptoms, 39 deficits) | accumulation of disability | Multivariate logistic regression model | Questionnaire: PA frequency [Exercise or hard physical ≥ 3 times/week vs. < 3 times/week], (2012) | OR 0.70 (0.70–0.80) | - | Age, sex, marital status, financial status, education level, physician visits |
| Savela et al. [21] 2013 Finland, Cohort Study | 514 | 48 | 47.5 (4.1), NM | - | 26 | A modification of Fried criteria (questionnaires: shrinking, weakness, exhaustion, low PA) | Physical frailty | Multinominal logistic regression models | Questionnaire: LTPA intensity [high vs. low] (1974) | OR 0.23 (0.08–0.65) | - | Age, BMI, smoking, blood pressure, and alcohol consumption in 1974, comorbidity in 2000 |
| Peterson et al. [24] 2009 USA, Cohort Study | 2964 | 323 | 73.6 (2.9), 70–79 | (51.0) | 5 | Gill frailty model (presence of functional limitations: gait speed < 0.60 m/s, or being unable to rise from a chair once with arms folded) | Physical frailty | Generalized estimating equation logistic regression models | Questionnaire: PA Intensity [Sedentary vs. Vigorous] (1997–1998) | OR 1.10 (0.75–1.63) | 0.91 (0.61–1.33) | Age, sex, race, education, marital status, smoking status, drinking status, waist circumference, and count of diagnoses |
| Abe et al. [25] 2020 Japan, Cohort Study | 2633 | 441 | 74.7 (6.5), ≥65 | 1367 (51.9) | 5 | the Kaigo-Yobo Checklist | Physical frailty | Multilevel multinomial logistic regression analysis | Questionnaire: PA frequency, [Exercise ≥ 3 vs < 3 days/week] (2012) | OR 0.77 (0.62–0.96) | - | Age, gender, BMI, subjective economic status, living arrangement, stroke, and diabetes |
| Niederstrasser et al. [26] 2019 England, Cohort Study | 7420 | 2441 | 66.9 (10.1), ≥50 | 3945 (53.2) | 12 | Frailty index (disease-related symptoms, self-reported conditions, activities of daily living, mobility, cognition, chronic diseases, or self-rated health, vision, and hearing) | multidimensional approach | Cox proportional hazards regression models | Questionnaire: LTPA volume, [vigorous vs. sedentary] (2004/2005) | HR 0.46 (0.36–0.57) | - | Sex, age, pain, PA level, wealth quintiles, educational qualifications, smoking, lower body strength, loneliness and BMI and waist-hip ratio |

*(Continued)*

**Table 1.** (Continued)

| Author, year, region | Total samples/, n | Frailty cases, n | Age, years, mean (SD), range | Female, n (%) | Follow-up, years, mean (SD) | Frailty assessment approach * | Frailty assessment approach classification | Statistical analysis | PA assessment method, PA indicator [comparison], evaluation year * | Effect estimator | Transformation of effect estimator by using the reciprocal of the original value † | Confounders |
|---|---|---|---|---|---|---|---|---|---|---|---|---|
| Kolehmainen et al. [27] 2020 Finland, Cohort study | 1137 | 319 | 56.0 (10.9), 60–77 | 522 (45.9) | 13.6 (10.2) | a modification of Fried criteria (shrinking, exhaustion, weakness, low PA, and slowness) | Physical frailty | Binomial logistic regression analysis | Questionnaire: LTPA volume [high vs. Low] (1972–2007) | OR 0.38 (0.23–0.61) | – | Age at earlier life assessment, sex, BMI, education, follow-up time, earlier life chronic diseases, smoking, research centre and older age diseases. |
| Yuki et al. [28] 2019 Japan Cohort study | 401 | 108 | 71.1 (4.3), 65–82 | 178 (44.4) | 4.2 (3.4) | A modification of the Cardiovascular Health Study criteria (shrinking, exhaustion, weakness, low PA, and slowness) | Physical frailty | Generalized estimate equation | The uniaxial accelerometry sensor: steps number [<5000 vs. ≥5000 steps/d]; LPA [<40.3 vs. ≥40.3 min/d]; MVPA [<7.5 vs. ≥7.5 min/d] (2000.4–2002.5) | OR 1.85 (1.10–3.11) OR 1.35 (0.82–2.25) OR 1.80 (1.05–3.09) | 0.54 (0.32–0.91) 0.74 (0.44–1.20) 0.56 (0.32–0.95) | Follow-up year, sex, age, body fat, education, current smoking, energy intake, number of comorbidities, and frailty status at baseline |
| Gil-Salcedo et al. [29] 2020 UK Cohort study | 6357 | 445 | 50.4 (2.1), ≥50 | 1856 (29.2) | 20.4 (5.9) | the Fried's frailty phenotype (shrinking, weakness, exhaustion, low PA, slowness) | Physical frailty | Competing risk model | Questionnaire: PA duration [active vs. inactive] (1985, 1991, 1997 and 2002) | HR 0.66 (0.48–0.88) | – | Sex, ethnicity, marital status, wave of inclusion, education, occupational position, the number of morbidities at age 50, all other health behaviors. |
| Yu et al. (1) [30] 2017 Hong Kong Cohort study | 4000 | 663 | 75.21 (6.73), ≥65 | 2000 (50.0) | 14 | The multiple deficits approach (self-reported medical and drug histories, functional assessments and psychological well-being, geriatric syndromes, 30 items) | multidimensional approach | Multiple logistic regression | Questionnaire: PA frequency [Low vs. high] (2001–2003) | OR 1.51 (1.27–1.81) | 0.66 (0.55–0.79) | Sex, age, education, smoking, alcohol use, living alone, Area Under the Curve |
| Yu et al. (2) [30] 2017 Taiwan urban, Cohort study | 963 | 317 | 75.23 (6.89), ≥65 | 440 (45.7) | 22 | The multiple deficits approach (self-reported medical and drug histories, functional assessments and psychological well-being, geriatric syndromes, 30 items) | multidimensional approach | Multiple logistic regression | Questionnaire: PA frequency [Low vs. high] (2003) | OR 2.03 (1.52–2.71) | 0.49 (0.37–0.66) | Sex, age, education, smoking, alcohol use, living alone, Area Under the Curve |
| Yu et al. (3) [30] 2017 Taiwan rural, Cohort study | 1429 | 538 | 75.38 (7.03), ≥65 | 687 (48.1) | 22 | The multiple deficits approach (self-reported medical and drug histories, functional assessments and psychological well-being, geriatric syndromes, 30 items) | multidimensional approach | Multiple logistic regression | Questionnaire: PA frequency [Low vs. high] (2003) | OR 2.29 (1.82–2.88) | 0.44 (0.35–0.55) | Sex, age, education, smoking, alcohol use, living alone, Area Under the Curve |

* Abbreviation: PA: physical activity; ADL: living of activity living; BMI: body mass index; LPA: light-intensity physical activity (1.8–3.0 metabolic equivalent of task, METs); LTPA: Leisure-Time Physical Activity; MVPA: moderate to vigorous intensity physical activity (≥3 METs).

†The transformation of effect estimator is the reciprocal the original value.

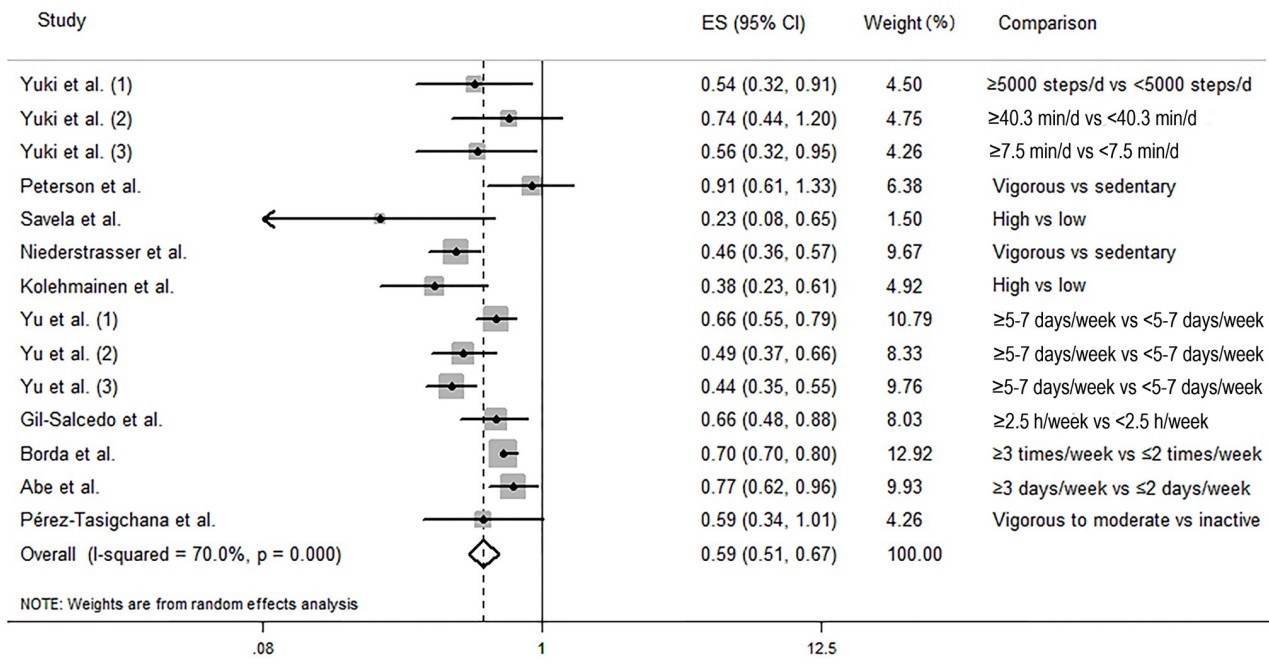

**Fig 2. Forest plot of association between physical activity and the risk of frailty.**

PA questionnaires for the old adults [19, 24], and the assessment of PA in other seven studies was with simple questions. Three studies assessed the PA frequency [20, 25, 30], three assessed the PA volume [19, 26, 27], two assessed the PA intensity [21, 24], one assessed the PA duration [29], and one assessed the steps and PA duration [28].

The random-effects model was used to calculate the pooled effect as the substantial heterogeneity was found ($I^2$ = 70.0%, $P$-heterogeneity < 0.001). As shown in Fig 2, in comparison to the participants with the lowest level of physical activity, those with the highest level of physical activity were significantly associated with a decreased odds of frailty (pooled ES: 0.59, 95% CI: 0.51–0.67). Visual inspection of funnels plots did not find obvious asymmetry (Fig 3). Begg's test ($P$ = 0.584) and Egger's test ($P$ = 0.067) did not reveal any significant publication bias. Sensitivity analysis was conducted 14 times by omitting one record each time and no significant change was observed in each analysis (S1 Fig).

When stratified by frailty assessment approach and compared with participants with the lowest PA level, participants within the highest level of PA were associated with 37% (pooled ES: 0.63, 95% CI: 0.52–0.77), 49% (pooled ES: 0.51, 95% CI: 0.41–0.63) and 30% (pooled ES: 0.70, 95% CI: 0.65–0.75) reduced odds of frailty when pooling studies using criteria of physical frailty, multidimensional model, and accumulation of disability, respectively (Table 2). When stratified by PA assessment tools, a similar protective effect was observed for studies using questionnaires (pooled ES: 0.58, 95% CI: 0.50–0.68) and using Uniaxial accelerometry sensor (pooled ES: 0.61, 95% CI: 0.45–0.83), respectively. Stratified analysis was also performed according to PA indicators, the region, sample size, female proportion, age, effect estimates, and follow-up, and a consistent protective effect was found in any subgroups (Table 2). The forest plot of association between PA and the odds of frailty by PA indicators was shown in S2 Fig. The meta-regression analysis was further conducted by putting the characteristics one by one into the meta-regression model separately, and the results showed that the follow-up period ($P$ = 0.012) contributed to the heterogeneity across the studies.

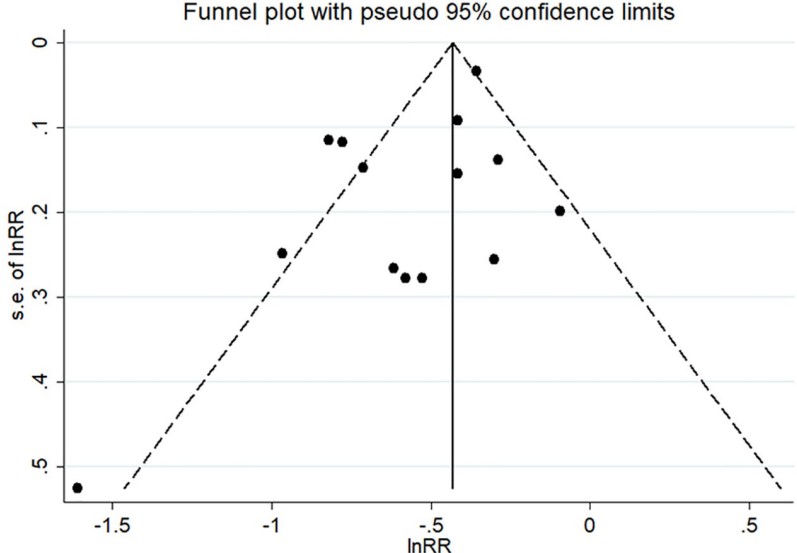

**Fig 3. Funnel plots with pseudo 95% confidence limits for the association between physical activity and the risk of frailty.**

Among the other characteristics, similarly no significant moderating effect was observed: region ($P = 0.967$), sample size ($P = 0.446$), female proportion ($P = 0.643$), age ($P = 0.204$), and frailty assessment approach ($P = 0.168$), PA measuring methods ($P = 0.842$), PA indicators ($P = 0.647$), and effect estimates ($P = 0.636$).

## Discussion

To the best of our knowledge, this study comprehensively summarized the overall effects of PA on frailty risk among the community-dwelling older residents. Based on available evidence with moderate quality involving 34,943 participants, results from this systematic review and meta-analysis suggest that a higher level of PA was significantly associated with lower odds of frailty.

This study by synthesizing 10 population-based cohort studies with 14 records showed that a higher level of physical activity was associated with 41% decreased odds of frailty (37% for physical frailty; 49% for multidimensional frailty). Sensitivity analysis, subgroup analysis, and meta-regression analysis yielded similar protective effects, indicating that our result was stable and robust. The result of our study was consistent with other reports, which showed that a higher level of physical activity was beneficial to physical function [22], muscle strength [39], and cognitive function [40]. In addition, similar protective effect of PA on the frailty was also reported by two cohort studies using linear regression model [41, 42]. Based on hourly accelerometry data, Huisingh-Scheetz et al. found that each frailty point corresponded a 7% lower mean hourly activity counts per minute among older adults by using mixed effects linear regression model [41]. Zhang et al. used multivariate linear regression models and showed that compared with participants with a continued regular PA frequency, participants with a decreased frequency were significantly more overall frailty [42].

A previous study tried to test the agreement between 35 different frailty assessment methods among the general population, and the results did not suggest to straightly pool or compare the prevalence or incidence yielded by diverse frailty scores. However, in our study, the

**Table 2. Overall and subgroup meta-analyses on the association between physical activity and the risk of frailty.**

| Subgroups | Number of studies/ records | Effect estimates (95% CI) | $P_h$ * | $I^2$ (%) | $P_B$† | $P_E$# |
|---|---|---|---|---|---|---|
| **All** | 14 | 0.59 (0.51–0.67) | < 0.001 | 70.0 | 0.584 | 0.067 |
| **Region** | | | | | | |
| Asia | 7 | 0.59 (0.49–0.71) | 0.013 | 62.7 | 0.764 | 0.773 |
| Europe and America | 7 | 0.58 (0.46–0.73) | 0.001 | 74.7 | 0.548 | 0.173 |
| **Sample size** | | | | | | |
| <1000 | 5 | 0.53 (0.42–0.66) | 0.347 | 10.4 | 0.806 | 0.694 |
| ≥1000 | 9 | 0.61 (0.52–0.71) | < 0.001 | 76.8 | 0.602 | 0.236 |
| **Female proportion** | | | | | | |
| <50% | 9 | 0.57 (0.48–0.68) | 0.001 | 68.3 | 0.466 | 0.047 |
| ≥50% | 5 | 0.59 (0.51–0.67) | 0.003 | 75.6 | 1.000 | 0.815 |
| **Age** | | | | | | |
| <70 | 5 | 0.68 (0.56–0.82) | 0.143 | 41.7 | 0.221 | 0.481 |
| ≥70 | 9 | 0.55 (0.46–0.66) | 0.003 | 65.1 | 0.917 | 0.617 |
| **Frailty assessment approach** | | | | | | |
| Physical Frailty | 9 | 0.63 (0.52–0.77) | 0.068 | 45.1 | 0.175 | 0.038 |
| Multidimensional model | 4 | 0.51 (0.41–0.63) | 0.019 | 69.9 | 1.000 | 0.288 |
| Accumulation of disability | 1 | 0.70 (0.65–0.75) | - - | - - | - - | - - |
| **Follow-up period** | | | | | | |
| <10 year | 7 | 0.70 (0.66–0.75) | 0.607 | 0.0 | 0.368 | 0.848 |
| ≥10 year | 7 | 0.51 (0.42–0.61) | 0.009 | 63.0 | 0.548 | 0.190 |
| **Physical activity (PA) indicators** | | | | | | |
| PA frequency | 5 | 0.61 (0.51–0.73) | < 0.001 | 81.0 | 0.221 | 0.261 |
| PA volume | 3 | 0.46 (0.38–0.56) | 0.498 | 0.0 | 1.000 | 0.914 |
| PA intensity | 2 | 0.50 (0.13–1.90) | 0.016 | 82.8 | 1.000 | - - |
| PA duration | 3 | 0.66 (0.52–0.83) | 0.760 | 0.0 | 1.000 | 0.872 |
| Step | 1 | 0.54 (0.32–0.91) | - - | - - | - - | - - |
| **PA assessment methods** | | | | | | |
| Questionnaire | 11 | 0.58 (0.50–0.68) | <0.001 | 76.3 | 0.533 | 0.079 |
| Uniaxial accelerometry sensor | 3 | 0.61 (0.45–0.83) | 0.646 | 0.0 | 1.000 | 0.396 |
| **Effect estimates** | | | | | | |
| Odds ratio | 12 | 0.60 (0.52–0.70) | <0.001 | 67.6 | 0.373 | 0.093 |
| Hazard ratio | 2 | 0.54 (0.38–0.77) | 0.063 | 71.1 | 1.000 | - - |

* P value for heterogeneity from Q-test

† P value from Begg's test

# P value from Egger's test

subgroup analysis and sensitivity analysis in our study showed the similar protective effect of PA on frailty, regardless of frailty evaluation approach (physical frailty, multidimensional approach, or accumulation of disability), suggesting that PA generated essential protective effect on frailty, no matter which frailty assessment approach was adopted.

PA is a multi-dimensional module, and therefore there is no measure that can assess all facets of PA [43]. Different PA assessments may reflect different facets, and this requires more attention when pooling the results from different studies. In our study, we did a series of repeated analyses 14 times by omitting one record each time and no significant change

was observed in each analysis. The stratified analysis by PA indicator also yielded a similar negative association of frailty with PA volume, PA frequency, PA intensity, and PA duration, respectively. Yuki et al. adopted a uniaxial accelerometry sensor to measure the number of steps and PA intensity [28], and other nine studies measure PA by using a questionnaire [19–21, 24–27, 29, 30]. However, similarly, we yielded a similar negative association of frailty with PA after excluding the study by Yuki et al [28]. We also tried to pool the three records reported in Yuki's study, and consistently we observed a protective effect by PA on frailty. These manifested that diversity in the PA measurement tools or indicators didn't influence the stability of the results. Multicomponent PA for older adults was recommended by WHO guidelines to maintain their physical fitness [44]. Consequently, no matter what type, intensity, frequency of PA, a higher level of PA in keeping moving and avoiding sedentary behaviours could yield benefits for the older adults.

Some studies showed that the duration of the follow-up period influenced the results from the longitudinal studies [45, 46]. Our study by synthesizing ten cohort studies found that the follow-up period may contribute to significant heterogeneity across the studies. However, our stratified analysis by follow-up period (<10 years vs. ≥10 years) yielded a similar beneficial effect by PA on frailty risk in both strata. One previous study [19] has also showed that a higher level of PA has a similar protective on the incidence of frailty after both longer-term and shorter-term follow-up. People in different countries had various exercise habits [47], but our stratified analysis by region (Asia vs. Europe and America) showed similar protective effects, indicating the effect of PA on frailty was not influenced by exercise habits. Age, gender, and sample size were the major confounders and varied in different studies. However, the stratified analysis in our study by sample size ($< 1000$ vs. $\geq 1000$), female proportion ($< 50\%$ vs. $\geq 50\%$) and, age ($< 70$ years vs. $\geq 70$ years) yielded the same results. These further showed the robustness of our results.

The mechanism underlying the association between PA and frailty is still unclear. However, PA has physiological effects on several inter-related physiological systems, and the latter can lead to frailty [48]. More PA can help people to control blood pressure and cholesterol, and therefore to reduce the risk of cardiovascular and metabolic diseases [49]. In skeletal muscle, PA can accelerate fatty acid oxidation, thus reducing the risk of cardiovascular diseases and type 2 diabetes mellitus [50, 51]. Regular exercise may exert characteristic changes in epigenetic mechanisms in skeletal muscle, specifically with respect to the genes associated with muscle growth and metabolism [52]. In the nervous system, keeping regular physical exercise helped to increase the blood flow to the brain and maintain the cognition function of older people [53, 54]; physical exercises seemed to maintain the longevity of motor neurons controlling the leg muscles [55, 56]. Previous researches showed that exercises can prevent older people in the community from falling [57], and even when a fall occurs, people who kept regular exercises were at a reduced risk to suffer from fractures because of the higher bone mineral density in their stronger bones [58].

Our study provides the basis for future research. First, the evidence on the associations of frailty and PA were mainly from the high-income countries rather than low-income or middle-income countries; the latter are experiencing a rapid ageing society, but the accessibility to health care resources is substantially shortage. The relevant studies in these low- or middle-income countries are encouraging. Second, instead of using a single question to assess the PA, the systematic and validified official PA questionnaire or accelerometers is encouraging to use in future studies to determine the dose-response relationship and determine the optimal exercise level for frailty prevention. Third, the definition of frailty urgently requires a consistent golden standard to facilitate the comparison between different studies. The development of tools to objectively measure frailty would help to achieve this goal.

One of the strengths of this study is that the studies included had moderate to high quality. Another strength is that all studies were cohort studies, hence the temporal sequence of causality is credible. Nevertheless, some potential limitations should be recognized in our studies. First, the frailty in the included studies was defined by different approaches, and PA was assessed by using different scales, these may result in information bias and thus may affect the consistency of the pooled results; however, the sensitivity and stratified analysis did not find any significant change. Then, the heterogeneity in some pooled analyses was significant, this may be due to the diversity of the characteristics of the subjects, the measurement of exposure, and the measurement of outcome in each study. For instance, the study conducted by Savela et al. was limited in Caucasian men of high socioeconomic status [21]; most of the included studies used self-report questionnaire to derive levels of PA, but the survey questions used in these studies varied largely. Hence, more studies with the same PA measurements and same frailty evaluation method are needed to further validate our results. Third, almost all the included studies assessed PA at the baseline, however, the longitudinally change of PA level and whether these changes influenced the incidence of frailty were unclear. This should be studied in the future. Fourth, because of the long follow-up in some studies, the missing data from those who died during follow-up may dilute the association between PA and frailty; therefore, more effective analyzed methods such as competing risk model should be considered.

In conclusion, this systematic review and meta-analysis with moderate-certainty evidence suggests that a higher level of PA was significantly related to decreased odds of frailty, and the benefits of PA for frailty prevention are independent of frailty assessment tools, PA indicators, and PA assessment methods. Findings from this study may help implement active exercise strategies to prevent frailty.

## Supporting information

**S1 Table. Search strategy of the study.**
(DOCX)

**S2 Table. Detailed quality assessment of the included studies by using criteria of the Newcastle-Ottawa Scale.**
(DOCX)

**S3 Table. Quality of evidence evaluated by GRADE approach.**
(DOCX)

**S1 Fig. Sensitivity analyses on the association between physical activity and the risk of frailty by excluding one study each time.**
(DOCX)

**S2 Fig. Forest plot of association between physical activity and the risk of frailty by physical activity indicators.**
(DOCX)

**S1 Checklist. PRISMA 2020 checklist.**
(DOCX)

## Acknowledgments

We would like to show our thanks to the authors of the originally published researches; their achievements are the prerequisites of our study.

## Author Contributions

**Conceptualization:** Akiko Tamakoshi, Xudong Liu.

**Data curation:** Peng Hu, Xudong Liu.

**Formal analysis:** Wenjing Zhao, Peng Hu.

**Funding acquisition:** Wenjing Zhao.

**Investigation:** Wenjing Zhao, Peng Hu, Weidi Sun.

**Writing – original draft:** Wenjing Zhao, Peng Hu.

**Writing – review & editing:** Weidong Wu, Jinhua Zhang, Hai Deng, Jun Huang, Shigekazu Ukawa, Jiahai Lu, Akiko Tamakoshi, Xudong Liu.

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
