## [Decision Letter · Decision Letter 0]

12 Apr 2022

PONE-D-22-03407Effect of physical activity on the risk of frailty: a systematic review and meta-analysisPLOS ONE

Dear Dr. Liu,

Thank you for submitting your manuscript to PLOS ONE. After careful consideration, we feel that it has merit but does not fully meet PLOS ONE’s publication criteria as it currently stands. Therefore, we invite you to submit a revised version of the manuscript that addresses the points raised during the review process.

We look forward to receiving your revised manuscript.

Kind regards,

Mohammad Meshbahur Rahman, MS.

Academic Editor

PLOS ONE

Journal Requirements:

(This study was supported by the Science and Technology Program of Guangzhou City (No.202102080404), the Guangdong Basic and Applied Basic Research Foundation (No. 2022A15150110X), the National Key R&D Program of China (No. 2018YFE0208000), and the Univers Foundation (No.17-02-160). The funders had no roles in the design, analysis, or writing of this manuscript.)

Please include your amended Funding Statement within your cover letter. We will change the online submission form on your behalf."

4. Ethics statement only appears at the end of the manuscript:

Your ethics statement should only appear in the Methods section of your manuscript. If your ethics statement is written in any section besides the Methods, please move it to the Methods section and delete it from any other section. Please ensure that your ethics statement is included in your manuscript, as the ethics statement entered into the online submission form will not be published alongside your manuscript. 

5. Please ensure that you refer to Figure 5 in your text as, if accepted, production will need this reference to link the reader to the figure.

6. Please include a caption for figure 5.

Additional Editor Comments: 

Please check author guidelines when submitting the revised manuscript.

Reviewers' comments:

Reviewer's Responses to Questions

**Comments to the Author**

1. Is the manuscript technically sound, and do the data support the conclusions?

Reviewer #1: Yes

Reviewer #2: Yes

2. Has the statistical analysis been performed appropriately and rigorously? 

Reviewer #1: Yes

Reviewer #2: Yes

3. Have the authors made all data underlying the findings in their manuscript fully available?

Reviewer #1: No

Reviewer #2: Yes

4. Is the manuscript presented in an intelligible fashion and written in standard English?

Reviewer #1: Yes

Reviewer #2: Yes

5. Review Comments to the Author

Reviewer #1: PONE-D-22-03407: This manuscript presents a systematic review and meta-analysis of Effect of physical activity on the risk of frailty. The authors conducted a systematic search, identifying 10 relevant studies. Study quality was evaluated using the Newcastle-Ottawa Scale (NOS). Meta-analysis was conducted the random-effects models.

The authors claim there was no systematic review and meta-analysis on the topic. But my search identifies 4 reviews. However, the systematic search demonstrates that there are enough studies to warrant a systematic review and meta-analysis. The quantitative analysis (including sensitivity analyses) appears to have been conducted carefully and appropriately, and the results should be of substantial interest to PLOS ONE readers and the environmental health community.

The manuscript omits several “best practices” that should be part of any systematic review, including preparation of a protocol, a clearly stated study question, and application of preferred approaches for evaluating and reporting risk of bias (or study quality, represented here by the NOS). The authors state that their review was conducted in accordance with the PRISMA guidelines, but several PRISMA recommendations have not been followed. An important challenge for this meta-analysis regardless of the endpoint, is that each study incorporates its own exposure contrast/defintion. The manuscript’s Discussion should address the interpretation of an effect estimate derived from meta-analysis of studies that differ in this important feature. It is also critical for the forest plots and tables to display the exposure contrast for each study. The manuscript contains multiple instances of inconsistent results reporting across the text, tables and figures; some specific instances are reported below.

Grammatical errors are found in places in the text.

Detailed Comments

The background should be improved by explaining the mechanism linking physical activity and frailty. There have been several systematic reviews and meta-analyses (e.g., de Vries NM, van et al. 2012;11:136–49, Theou et al 2011, Puts et a; 2017, etc and more recently a protocol by Pan et al 2019). The authors should discuss these reviews and articulate why there is the need for another review. The gap in these previous reviews should be articulated

L3/4: The objective should be stated in accordance with Population Exposure Comparator Outcome statement

L7: This is incorrect. The definition of physical activity varied widely among the studies. In some studies, it was defined as Vigorous/moderate physical activity, or exercise, or hard physical activity

>3 times/week, etc. How can the authors have derived highest and lowest PA.

L29: insert 'a' between 'at' and 'speed'

L32” delete pls ‘the’

L#36 and L#37 are the same-repetition.

L40: pls change 'preventive effect' to 'impact'

In the last sentence of the introduction, the objective of the systematic review was provided. It would be more clearer, and in accordance with recommendations by the PRISMA Statement (BMJ, 2009; 339:b2700), to identify some specific question that the systematic review is intended to address. The questions might include the populations, exposure, comparator, and outcomes of interest. These questions would also support the criteria and decisions made in literature search terms and screening/ selection of studies. The reason for restricting the studies to follow-up design should be clearly articulated in the background.

Search strategy

The PRISMA guideline recommends registration of a systematic review. The PRISMA checklist (#8) recommends presentation of the full electronic search strategy for at least one database – not provided in this manuscript.

The search strategy should be structured according to population, exposure comparator, outcome and study design.

The Eligibility criteria did not follow the PECO (S) statement. It should also be structured in accordance with Population, Exposure, Comparator, outcome (s) and study design (PECOS) statement. When stating the inclusion criteria (for example for Population) should also state the reasons for exclusion. Most of what the authors have there as inclusion and exclusion criteria are not; in accordance with the PRISMA's recommendation. For example, how can effect estimate be an inclusion/exclusion criteria. Are the authors not encouraging publication bias. This is a systematic review and a meta-analysis (which also both quantitative and qualitative reporting).

L67: With the PA types specified, the authors should give examples.

L70-71: If the authors had followed the PRISMA recommendations closely. some of what they have here as exclusion criteria would not be here. For example, animal studies, reviews, trials, letters, abstracts, etc would have been removed with search filters. Please, delete this. You did not apply your inclusion/exclusion criteria before excluding them

The authors who did the data extraction and assessment should be identified

L77-80: Newcastle-Ottawa Scale and scoring of studies: I have concerns about the use of an overall scale or score to denote study “quality.” Moreover, the developers of the PRISMA Statement do not recommend this. Some of the evaluation components may have greater impact on determining the value of a particular study for the systematic review. Others, in particular the evaluation of confounding, need to be interpreted for the direction the bias is imposing on the risk estimate (e.g., away from or toward the null). That said, it appears that the NOS scale was used to group studies in sensitivity analyses, rather than to exclude studies from the review. This is an appropriate use of scoring, although I would not know how to interpret the scale of 7 out of 9 in terms of what it really means for “quality.”

The authors should provide detailed explanation of how the NOS was applied. On comparability what are the major and minor confounders considered. A table of NOS assessment including detailed consideration for comparability for each study should be provided in the appendix.

L82: Some of the studies reported OR, others reported RR and some also reported HR. RR, HR are approximated to OR only when OR<10%. The outcome of interest is not a rare outcome and why RR=HR=OR assumption in this study. There are other ways of converting OR to HR or RR and the vice versa.

L87: I realized that the authors reported RR in the tables and ES (effect estimates) in the forest plots. Please, change all RR to ES because the measure of effect were not consistent.

In Table 1 show exposure contrast for PA. This should also be done for the plots.

L100: The flow chart was a bit messy.

For example, 0 articles identified through other sources. What are those other sources? pls delete.

Among the 6424 articles were RCTS you excluded. Are these RCTs on PA and fraility? If yes why delete them because they are also longitudinal designs.

I strongly recommend that the 8 studies that did not provide OR/HR/RR and were excluded should be included in your qualitative analysis.

L107: Provide an exposure contrast for physical activity in Table 1 and this should also be shown in the forest plots.

L130-132: The included studies varied widely on the way in which fraility and PA were measured. For example they reported PA frequency, or PA volum or PA intensity and PA duration. According to the authors PA is a multi-dimensional module and different PA assessments may reflect different facets. I do think the authors need to report overall summary effects. I recommend forest plot should reflect these differences in exposure definition and (summary estimates for each definition) also show PA contrast (e.g., low vs high).

I think the results can be presented descriptively without going through the meta-analysis, subgroup analysis and meta-regresssion

L136-137: The authors should know that funnel visualization and the Beggs method do not perform well with few studies.

L185-187: This is the more reason why i think, pooling the results is not a good idea. Analysis could be done for each PA definition as suggested above. Again gave the fact that the studies are not many.

L23-224: The authors should also discuss inherent limitations in included studies.

Other limitations in the study include the assessment of quality as against the risk of bias.

Reviewer #2: Thank you for the opportunity to review this systematic review investigating the relationship between physical activity (PA) and the risk of frailty. The topic is relevant to public health and clinicians, and the findings have the potential to contribute to the field by providing pooled estimates of the association between PA and frailty risk from cohort studies. I have a few comments and suggestions:

Introduction - Page 3 line 37: Please remove the duplicate sentence: "such as the musculoskeletal, cardiorespiratory, endocrine, and nervous systems"

Methods - Page 4 line 66: I have the impression the inclusion criteria could be improved and more detailed. Could you please use the PICO format to assist readers understanding the type of study included? I believe it is unclear how frailty outcomes were considered in this review. Also, there is no information about the population (please add the age range, if applicable). Apologies if I missed it.

Results

Table 2. The title suggests subgroup analyses only, but I think you also included the overall analyses. Please review this to inform readers table 2 includes overall and subgroup meta-analyses. Also, it is difficult to quickly identify significant poolings. Could you please bold the significant results?

Confounders. Please add information about confounders. I noticed you added a sentence in your discussion, but this may not be enough.

Adjusted analysis. Please add information on whether you included unadjusted and adjusted results in the meta-analysis. I have the impression this is unclear in your manuscript. Moreover, could you conduct metaanalysis including only adjusted analysis?

Page 7 line 133. I find it interesting that you described moderate heterogeneity for your main metaanalysis. I believe this is more likely to be classified as substantial heterogeneity (following the Cochrane Handbook). Please justify why a heterogeneity of 70% should be considered moderate in your review or reclassify it.

Suggestion: To strengthen your review, could you include the GRADE approach to summarise the certainty of evidence?

I hope some of these comments are helpful.

Thank you for conducting this study

6. PLOS authors have the option to publish the peer review history of their article (what does this mean?). If published, this will include your full peer review and any attached files.

Reviewer #1: **Yes: **Dr. Reginald Quansah

Reviewer #2: No

---

## [Author Response · Author response to Decision Letter 0]

12 Jun 2022

Re: Effect of physical activity on the risk of frailty: a systematic review and meta-analysis (PONE-D-22-03407)

We thank the reviewers and editors for his/her valuable comments, our point-by-point responses are shown below the comments. The changes made in the revised manuscript were highlighted in red.

C = comment; R = reply

#To reviewer 1

C1: The authors claim there was no systematic review and meta-analysis on the topic. But my search identifies 4 reviews. However, the systematic search demonstrates that there are enough studies to warrant a systematic review and meta-analysis. The quantitative analysis (including sensitivity analyses) appears to have been conducted carefully and appropriately, and the results should be of substantial interest to PLOS ONE readers and the environmental health community. 

R1: Thanks for reviewer’s comment. We appreciate the reviewers for the recognition of our quantitative analysis and the significance of the results. We have carefully updated literature in the manuscript accordingly:

“Physical activity (PA) can regulate the function of multiple systems in the body [1]; the dysfunction of these body system can further contribute to the development of frailty [1]. A systematic review [15], a scoping review [16], and a systematic review and meta-analysis [17] synthesized the evidence and found that exercise training can reduce the level of frailty and improve the prognosis of frailty among older adults. However, epidemiological studies on the relationship between PA and frailty did not reach to a consistent conclusion. For instance, Trombetti [18] and Pérez-Tasigchana [19] did not find any association between PA and frailty, while Borda [20] and Savela [21] reported that PA was a protective factor for frailty. A meta-analysis found that physical exercise therapy could improve mobility and physical functioning in elderly patients suffering from mobility problems, disability and/or multi-morbidity [22]; however, more original studies have been reported in the following decade, making it necessary to update the synthetic evidence. A more recent systematic review including four randomized controlled trials and two prospective cohort studies found that physical activity might be an effective intervention for preventing frailty mong people aged 65 years and older [23]; nevertheless, the intervention measures varied largely among four RCTs and one included RCT adopted combined nutrition and exercise interventions, limiting the application of the results. Moreover, this study[23] only included two cohort studies, but more cohort studies have been reported [18-21,24-30]. In addition, there are many evaluation methods for frailty, and no unified standard has been formed [31]. Whether different methods of frailty assessment influence the association between PA and frailty onset is unclear.” [lines 34-52]

C2: The manuscript omits several “best practices” that should be part of any systematic review, including preparation of a protocol, a clearly stated study question, and application of preferred approaches for evaluating and reporting risk of bias (or study quality, represented here by the NOS). The authors state that their review was conducted in accordance with the PRISMA guidelines, but several PRISMA recommendations have not been followed. An important challenge for this meta-analysis regardless of the endpoint, is that each study incorporates its own exposure contrast/defintion. The manuscript’s Discussion should address the interpretation of an effect estimate derived from meta-analysis of studies that differ in this important feature. It is also critical for the forest plots and tables to display the exposure contrast for each study. The manuscript contains multiple instances of inconsistent results reporting across the text, tables and figures; some specific instances are reported below.

R2: Thanks for valuable comments. The whole manuscript has carefully revised accordingly. 

C3: Grammatical errors are found in places in the text. 

R3: Thanks for comments. We have checked the gramma and revised the manuscript.

C4: The background should be improved by explaining the mechanism linking physical activity and frailty. There have been several systematic reviews and meta-analyses (e.g., de Vries NM, van et al. 2012;11:136–49, Theou et al 2011, Puts et a; 2017, etc and more recently a protocol by Pan et al 2019). The authors should discuss these reviews and articulate why there is the need for another review. The gap in these previous reviews should be articulated.

R4: Thanks for the good suggestion. The background in the introduction has been updated as follows:

“Physical activity (PA) can regulate the function of multiple systems in the body [1]; the dysfunction of these body system can further contribute to the development of frailty [1]. A systematic review [15], a scoping review [16], and a systematic review and meta-analysis [17] synthesized the evidence and found that exercise training can reduce the level of frailty and improve the prognosis of frailty among older adults. However, epidemiological studies on the relationship between PA and frailty did not reach to a consistent conclusion. For instance, Trombetti [18] and Pérez-Tasigchana [19] did not find any association between PA and frailty, while Borda [20] and Savela [21] reported that PA was a protective factor for frailty. A meta-analysis found that physical exercise therapy could improve mobility and physical functioning in elderly patients suffering from mobility problems, disability and/or multi-morbidity [22]; however, more original studies have been reported in the following decade, making it necessary to update the synthetic evidence. A more recent systematic review including four randomized controlled trials and two prospective cohort studies found that physical activity might be an effective intervention for preventing frailty mong people aged 65 years and older [23]; nevertheless, the intervention measures varied largely among four RCTs and one included RCT adopted combined nutrition and exercise interventions, limiting the application of the results. Moreover, this study[23] only included two cohort studies, but more cohort studies have been reported [18-21,24-30]. In addition, there are many evaluation methods for frailty, and no unified standard has been formed [31]. Whether different methods of frailty assessment influence the association between PA and frailty onset is unclear. 

Pan and colleagues published a registered protocol for systematic review and meta-analysis to demonstrate the association between PA and the risk of frailty in the old community-dwelling residents [32]; however, the related systematic review and meta-analysis has not been reported in the past three years. What’s more, this protocol [32] did not mention how to define frailty and whether synthesize the evidence by frailty definition or not. 

Therefore, we conducted this systematic review and meta-analysis by synthesize evidence from cohort studies to examine the effect of PA on the onset of frailty among the community-dwelling middle and older age adults, so to provide evidence for frailty prevention for middle and order age adults.” [Line 34-61]

C5: L3/4: The objective should be stated in accordance with Population Exposure Comparator Outcome statement

R5: Thanks for your comment, we updated the objective in the abstract and in the introduction as follows: 

“This systematic review with meta-analysis aimed to evaluate the effect of PA on the onset of frailty in the community-dwelling middle and order age adults by pooling data from cohort studies.” [Lines 3-5]

“Therefore, we conducted this systematic review and meta-analysis by synthesize evidence from cohort studies to examine the effect of PA on the onset of frailty among the community-dwelling middle and older age adults, so to provide evidence for frailty prevention for middle and order age adults.” [Lines 58-61]

C6: L7: This is incorrect. The definition of physical activity varied widely among the studies. In some studies, it was defined as Vigorous/moderate physical activity, or exercise, or hard physical activity>3 times/week, etc. How can the authors have derived highest and lowest PA.

R6: Thank you for the comment. The definition of physical activity indeed varied widely in these studies. However, in these studies, the PA indicator often was transformed to categorical variable, and then the effect of PA on the frailty risk was estimated by comparing the high category with low category (such as high vs. Low, vigorous vs. sedentary), or by comparing the low category with high category (such as Sedentary vs. Vigorous). To facilitate the pooled analysis, we transformed the comparison of “low category with high category” to the comparison of “high category with low category” by using the reciprocal method suggested by Cooper and colleague [Ref.36 : Cooper HM, Hedges LV, Valentine JC (2019) The handbook of research synthesis and meta-analysis. New York: Russell Sage Foundation]. 

Now we have updated the table 1 and figure 2, adding the comparison of PA levels.

C7: L29: insert 'a' between 'at' and 'speed'

R7: Thanks for your comment, we have inserted “a” accordingly. [Line 30]

C8: L32” delete pls ‘the’

R8：Thanks for your comment, “the” has been deleted accordingly. [Line 33]

C9: L#36 and L#37 are the same-repetition.

R9：Thanks for raising this question, we have revised the paragraph and deleted the repeated content. [Line35]

C10: L40: pls change 'preventive effect' to 'impact'

R10: Thanks for your comment, the paragraph in which the phrase “preventive effect” was placed has been revised. 

C11: In the last sentence of the introduction, the objective of the systematic review was provided. It would be more clearer, and in accordance with recommendations by the PRISMA Statement (BMJ, 2009; 339:b2700), to identify some specific question that the systematic review is intended to address. The questions might include the populations, exposure, comparator, and outcomes of interest. These questions would also support the criteria and decisions made in literature search terms and screening/ selection of studies. The reason for restricting the studies to follow-up design should be clearly articulated in the background.

R11: Thanks for suggestion. We have revised accordingly as follows: 

“Physical activity (PA) can regulate the function of multiple systems in the body [1]; the dysfunction of these body system can further contribute to the development of frailty [1]. A systematic review [15], a scoping review [16], and a systematic review and meta-analysis [17] synthesized the evidence and found that exercise training can reduce the level of frailty and improve the prognosis of frailty among older adults. However, epidemiological studies on the relationship between PA and frailty did not reach to a consistent conclusion. For instance, Trombetti [18] and Pérez-Tasigchana [19] did not find any association between PA and frailty, while Borda [20] and Savela [21] reported that PA was a protective factor for frailty. A meta-analysis found that physical exercise therapy could improve mobility and physical functioning in elderly patients suffering from mobility problems, disability and/or multi-morbidity [22]; however, more original studies have been reported in the following decade, making it necessary to update the synthetic evidence. A more recent systematic review including four randomized controlled trials and two prospective cohort studies found that physical activity might be an effective intervention for preventing frailty mong people aged 65 years and older [23]; nevertheless, the intervention measures varied largely among four RCTs and one included RCT adopted combined nutrition and exercise interventions, limiting the application of the results. Moreover, this study [23] only included two cohort studies, but more cohort studies have been reported [18-21,24-30]. In addition, there are many evaluation methods for frailty, and no unified standard has been formed [31]. Whether different methods of frailty assessment influence the association between PA and frailty onset is unclear. 

Pan and colleagues published a registered protocol for systematic review and meta-analysis to demonstrate the association between PA and the risk of frailty in the old community-dwelling residents [32]; however, the related systematic review and meta-analysis has not been reported in the past three years. What’s more, this protocol [32] did not mention how to define frailty and whether synthesize the evidence by frailty definition or not. 

Therefore, we conducted this systematic review and meta-analysis by synthesize evidence from cohort studies to examine the effect of PA on the onset of frailty among the community-dwelling middle and older age adults, so to provide evidence for frailty prevention for middle and order age adults.” [Line34-61]

C12: The PRISMA guideline recommends registration of a systematic review. The PRISMA checklist (#8) recommends presentation of the full electronic search strategy for at least one database – not provided in this manuscript. The search strategy should be structured according to population, exposure comparator, outcome and study design. The Eligibility criteria did not follow the PECO (S) statement. It should also be structured in accordance with Population, Exposure, Comparator, outcome (s) and study design (PECOS) statement. When stating the inclusion criteria (for example for Population) should also state the reasons for exclusion. Most of what the authors have there as inclusion and exclusion criteria are not; in accordance with the PRISMA's recommendation. For example, how can effect estimate be an inclusion/exclusion criteria. Are the authors not encouraging publication bias. This is a systematic review and a meta-analysis (which also both quantitative and qualitative reporting).

R12: Thanks for reviewer’s comments. The PRISMA guideline recommends registration of a systematic review; however, it is not a mandatory component. In fact, many systematic reviews published even in high-prestigious journals (such as JAMA, Lancet, JACC) also did not with a registration.

As we stated in the manuscript, we searched literatures from three datasets (PubMed, Embase, and Web of science). We used the keywords and retrieval strategy as follows: “(‘physical activity’ OR ‘exercise’ OR ‘acute exercise’ OR ‘isometric exercises’ OR ‘aerobic exercise’ OR ‘physical exercise’ OR ‘endurance exercise’ OR ‘resistance exercise’ OR ‘strength exercises’ OR ‘training’ OR ‘exercise training’, OR ‘combined training’, OR ‘weight-lifting’ OR ‘running’ OR ‘jogging’ OR ‘swimming’ OR ‘walking’ OR ‘yoga’ OR ‘Tai chi’ OR ‘daily activity’ OR ‘lifestyle’ OR ‘sport’) AND (‘frailty’ OR ‘frailness’ OR ‘frailty syndrome’ OR and ‘debility’)”. We displayed the detailed search strategy in the supplementary Table S1. We also did a reverse reference citation tracking to find potential reports. The flow-chart of literature selection can be seen in Figure 1. [Lines 64-75, supplementary table S1]

We sincerely pay more attention to the publication bias. The misunderstanding might due to the incorrect description. To make clear for readers, we have revised the paragraph related the inclusion and exclusion criteria accordingly: “Inclusion criteria were as follows: the exposure of PA including exposure intensity, frequency, duration, volume, step, or any specific type (such as leisure-time physical activities, occupational activities, and exercise, etc.) was reported; frailty and its assessment approaches (physical frailty, multidimensional approach, accumulation of disability, etc.) were reported ; design of cohort studies; middle and order age healthy adults; the relationship between PA and frailty was evaluated; the study was published in English. Cross-sectional studies, animal studies, trials, reviews, editorials, letters, abstracts, and studies lacking data to manifest the relationship between PA and frailty were excluded after reviewing title, abstract or full-text.” [Lines 76-83]

C13: L67: With the PA types specified, the authors should give examples.

R13: Thanks for suggestion. The examples of specific PA types were given as follows: “(such as leisure-time physical activities, occupational activities, and exercise, etc.)” [Line: 77-78]

C14: L70-71: If the authors had followed the PRISMA recommendations closely. some of what they have here as exclusion criteria would not be here. For example, animal studies, reviews, trials, letters, abstracts, etc would have been removed with search filters. Please, delete this. You did not apply your inclusion/exclusion criteria before excluding them

R14: Thanks for your comments. W have updated the inclusion and exclusion criteria as follows: “Inclusion criteria were as follows: the exposure of PA including exposure intensity, frequency, duration, volume, step, or any specific type (such as leisure-time physical activities, occupational activities, and exercise, etc.) was reported; frailty and its assessment approaches (physical frailty, multidimensional approach, accumulation of disability, etc.) were reported ; design of cohort studies; middle and order age healthy adults; the relationship between PA and frailty was evaluated; the study was published in English. Cross-sectional studies, animal studies, trials, reviews, editorials, letters, abstracts, and studies lacking data to manifest the relationship between PA and frailty were excluded after reviewing title, abstract or full-text” [Lines: 76-83]

C15: The authors who did the data extraction and assessment should be identified

R15: Thanks for your comments, we have added such information in the section of authors’ contributions: “Data extraction and assessment: PH, WJZ & WDS”. [Lines: 284]

C16: L77-80: Newcastle-Ottawa Scale and scoring of studies: I have concerns about the use of an overall scale or score to denote study “quality.” Moreover, the developers of the PRISMA Statement do not recommend this. Some of the evaluation components may have greater impact on determining the value of a particular study for the systematic review. Others, in particular the evaluation of confounding, need to be interpreted for the direction the bias is imposing on the risk estimate (e.g., away from or toward the null). That said, it appears that the NOS scale was used to group studies in sensitivity analyses, rather than to exclude studies from the review. This is an appropriate use of scoring, although I would not know how to interpret the scale of 7 out of 9 in terms of what it really means for “quality.” The authors should provide detailed explanation of how the NOS was applied. On comparability what are the major and minor confounders considered. A table of NOS assessment including detailed consideration for comparability for each study should be provided in the appendix.

R16: Thanks for your comments. In our manuscript, we provided the quality assessment results by using NOS (supplementary table S2). Now, we updated it and provided a more detailed table. Also, the detailed explanation was provided. [see supplementary table S2] 

C17: L82: Some of the studies reported OR, others reported RR and some also reported HR. RR, HR are approximated to OR only when OR<10%. The outcome of interest is not a rare outcome and why RR=HR=OR assumption in this study. There are other ways of converting OR to HR or RR and the vice versa.

R17: Thanks for your comments. Considering that only 2 records reported HR, and other 12 records reported OR, we pooled the results using effect estimates (ES) across the manuscript. Moreover, we further conducted subgroup analysis by effect estimates, and similar protective effect was observed for the studies using OR (pool ES: 0.60, 95% CI: 0.52 - 0.70) and the studies using HR (pool ES: 0.54, 95% CI: 0.38 - 0.77). We also updated all the results across the manuscript. [see Table 2]

C18: L87: I realized that the authors reported RR in the tables and ES (effect estimates) in the forest plots. Please, change all RR to ES because the measure of effect were not consistent.

R18: Thanks for comments, we changed the RR with ES across the manuscript.

C19: In Table 1 show exposure contrast for PA. This should also be done for the plots.

R19: Thanks for comments, we added the exposure contrast for PA in the forest plots. [see table S2, Figure 2]

C20: L100: The flow chart was a bit messy. For example, 0 articles identified through other sources. What are those other sources? pls delete.

R20: Thanks for your comments. We updated the flow-chart accordingly. [see Figure 1]

C21: Among the 6424 articles were RCTS you excluded. Are these RCTs on PA and frailty? If yes why delete them because they are also longitudinal designs.

R21: Thanks for your comments. Among the RCTs we excluded, most of the RCTs were not relevant to the topic of this study. Two RCTs (de Souto Barreto et al., 2018; Li et al., 2010) conducted with multidomain intervention with frailty; however, PA is a subitem of the intervention and the effects of PA were not reported or can be calculated. Nagai et al (Nagai et al., 2018) conducted a RCT to explore PA combined with resistance training on frailty symptoms; however, the study just reported the difference between groups and cannot estimate the effect size of each single component. Given the limited available evidence from RCT study and the target to ascertain the effect of PA on frailty incidence, we limited the included study to prospective cohort study to prevent the potential bias from different study designs. In fact, in the inclusion criteria of this study, we also limited the included study to cohort study.

C22: I strongly recommend that the 8 studies that did not provide OR/HR/RR and were excluded should be included in your qualitative analysis. 

R22: Thanks for your comments. Among 8 studies we excluded, 5 studies (Cheung et al., 2020; Ye et al., 2020; Lorenzo-López et al., 2019; Pao et al., 2018; Rogers et al., 2017) focused on the changes in frailty or the trajectories of frailty, which did not report the association between PA and frailty. One study from US explored the relationship between physical activity and frailty among older adults based on hourly accelerometry data, and the study showed that each frailty point corresponded a 7% lower mean hourly activity counts per minute by using mixed effects linear regression model (Huisingh-Scheetz et al., 2017). One study from Spain used multivariate linear regression models and showed that compared with participants with a continued regular PA frequency, participants with a decreased frequency were significantly more overall frailty (β = 1.31; 95% confidence interval = 0.99-1.63) (Zhang et al., 2020). One study conducted among 604 Japanese women showed that high exercise and high dietary varieties group was associated with lower risk of frailty (OR, 0.38, 95% CI, 0.15-0.92); however, this study did not report the effect between PA and frailty solely (OSUKA et al., 2019). 

Given to the limited evidence on PA and frailty left, we added the following sentences in the discussion section: “In addition, similar protective effect of PA on the frailty was also reported by two cohort studies using linear regression model [40,41]. Based on hourly accelerometry data, Huisingh-Scheetz et al. found that each frailty point corresponded a 7% lower mean hourly activity counts per minute among older adults by using mixed effects linear regression model [40]. Zhang et al. used multivariate linear regression models and showed that compared with participants with a continued regular PA frequency, participants with a decreased frequency were significantly more overall frailty [41].” [Line 197-203, see figure 1]

C23: L107: Provide an exposure contrast for physical activity in Table 1 and this should also be shown in the forest plots.

R23: Thanks for comments, we added such information in Table 1, Figure 2, and supplementary Figure S2.

C24: L130-132: The included studies varied widely on the way in which frailty and PA were measured. For example, they reported PA frequency, PA volume or PA intensity and PA duration. According to the authors PA is a multi-dimensional module and different PA assessments may reflect different facets. I do think the authors need to report overall summary effects. I recommend forest plot should reflect these differences in exposure definition and (summary estimates for each definition) also show PA contrast (e.g., low vs high). 

R24: Thanks for comments. We have provided the pooled results from all included studies and each PA indicators in table 2. Now we also added the forest plots by PA indicators in supplementary materials (see Supplementary Figure S2) and added the following sentence to the results section: “The forest plot of association between PA and frailty risk by PA indicators was shown in Figure S2.” [Lines 179-180]

C25: I think the results can be presented descriptively without going through the meta-analysis, subgroup analysis and meta-regression

R25: Thanks for the comments. In fact, almost the generalizability of the included studies was limited in specific regions. Hence, from the perspective of public health and guiding practice, we remain our view that it is necessary to conduct a pooled analysis and provide the synthetic results. Therefore, we conducted meta-analysis and subgroup analysis, drew forest plot of the relationship between PA and fragility risk and forest plot grouped according to PA indicators. We also did the meta-regression was conducted to assess the potential sources of heterogeneity.

C26: L136-137: The authors should know that funnel visualization and the Begger’s method do not perform well with few studies.

R26: Thanks for comments. Considering that only 14 records from ten studies were included, and the regression method is more sensitive than the rank correlation approach (Sterne et al., 2001), we conducted funnel plot together with Begger’s test, and Egger’s test to test the publication bias. The p-values from Begger’s test and Egger’s test were shown in Table 2.

C27: L185-187: This is the more reason why i think, pooling the results is not a good idea. Analysis could be done for each PA definition as suggested above. Again gave the fact that the studies are not many.

R27: Thanks for comments. In fact, we also did the analysis by each PA indicator, we added the results to table 2 and the forest plot to supplementary materials. [See Table 2 and Supplementary Figure S2].

C28: L23-224: The authors should also discuss inherent limitations in included studies. Other limitations in the study include the assessment of quality as against the risk of bias.

R28: Thanks for comments. The discussion has been updated by including the discussion of the limitations in included studies.

“Nevertheless, some potential limitations should be recognized in our studies. First, the frailty in the included studies was defined by different approaches, and PA was assessed by using different scales, these may result in information bias and thus may affect the consistency of the pooled results; however, the sensitivity and stratified analysis did not find any significant change. Then, the heterogeneity in some pooled analyses was significant, this may be due to the diversity of the characteristics of the subjects, the measurement of exposure, and the measurement of outcome in each study. For instance, the study conducted by Savela et al. was limited in Caucasian men of high socioeconomic status [21]; most of the included studies used self-report questionnaire to derive levels of PA, but the survey questions used in these studies varied largely. Hence, more studies with the same PA measurements and same frailty evaluation method are needed to further validate our results. Third, almost all the included studies assessed PA at the baseline, however, the longitudinally change of PA level and whether these changes influenced the incidence of frailty were unclear. This should be studied in the future. Fourth, because of the long follow-up in some studies, the missing data from those who died during follow-up may dilute the association between PA and frailty; therefore, more effective analyzed methods such as competing risk model should be considered.” [Lines 262-276]

#To reviewer 2

C1: Introduction - Page 3 line 37: Please remove the duplicate sentence: "such as the musculoskeletal, cardiorespiratory, endocrine, and nervous systems"

R1: Thanks for your suggestion, we have revised the paragraph. [Lines 34-52]

C2：Methods - Page 4 line 66: I have the impression the inclusion criteria could be improved and more detailed. Could you please use the PICO format to assist readers understanding the type of study included? I believe it is unclear how frailty outcomes were considered in this review. Also, there is no information about the population (please add the age range, if applicable). Apologies if I missed it.

R2: Thanks for your suggestion. We added the age range to the fourth column in table 1. We also updated the description of inclusion and exclusion criteria accordingly.

“Inclusion criteria were as follows: the exposure of PA including exposure intensity, frequency, duration, volume, step, or any specific type (such as leisure-time physical activities, occupational activities, and exercise, etc.) was reported; frailty and its assessment approaches (physical frailty, multidimensional approach, accumulation of disability, etc.) were reported ; design of cohort studies; middle and order age healthy adults; the relationship between PA and frailty was evaluated; the study was published in English. Cross-sectional studies, animal studies, trials, reviews, editorials, letters, abstracts, and studies lacking data to manifest the relationship between PA and frailty were excluded after reviewing title, abstract or full-text.” [Lines 76-83]

C3: Table 2. The title suggests subgroup analyses only, but I think you also included the overall analyses. Please review this to inform readers table 2 includes overall and subgroup meta-analyses. Also, it is difficult to quickly identify significant poolings. Could you please bold the significant results?

R3: Thanks for the good suggestion. We revised the title of Table 2 and we also bolded the significant results in table 2. 

C4: Confounders. Please add information about confounders. I noticed you added a sentence in your discussion, but this may not be enough.

R4: Thanks for the suggestion. In fact, we have showed the confounders of each included study in the general characteristics table [Table 1].

C5: Adjusted analysis. Please add information on whether you included unadjusted and adjusted results in the meta-analysis. I have the impression this is unclear in your manuscript. Moreover, could you conduct meta-analysis including only adjusted analysis?

R5: Thanks for comments. Considering that the results of the included studies would be more reliable after adjusting for potential confounders than the unadjusted results, in this meta-analysis, we evaluated the pooled effects based on the adjusted results in each included study.

We added such information in section of statistical analysis: “A meta-analysis was conducted to estimate the overall pooled effect estimates (ES) based on the adjusted odds ratio or hazard ratio and 95% confidence interval (CI), by comparing the highest with the lowest levels of PA.” [Line 105-107]

C6: Page 7 line 133. I find it interesting that you described moderate heterogeneity for your main meta-analysis. I believe this is more likely to be classified as substantial heterogeneity (following the Cochrane Handbook). Please justify why a heterogeneity of 70% should be considered moderate in your review or reclassify it.

R6: Thanks for comments. We rechecked the classification of heterogeneity according to the Cochrane Handbook, and found that described a heterogeneity of 70% as moderate heterogeneity was inaccurate. We correct it according to your suggestions.

“The random-effects model was used to calculate the pooled effect as the substantial heterogeneity was found (I2 = 70.0%, P-heterogeneity < 0.001).” [Line 162-163]

C7: Suggestion: To strengthen your review, could you include the GRADE approach to summarise the certainty of evidence?

R7: Thanks for your suggestions. We further included the GRADE approach to evaluate the quality of the evidence. We found that the quality of the evidence from the outcomes evaluated by the GRADE system was classified as moderate (showed in Supplementary Table S3), and substantial heterogeneity was the main reason responsible for the limited quality of the evidence.

We accordingly added the following sentence to the section of Methods: “The Grading of Recommendations Assessment, Development and Evaluation (GRADE) approach was used to evaluated the quality of the body of evidence [29]. The GRADE approach was based on considerations such as study design, risk of bias, inconsistency, imprecision, indirectness, publication bias and other aspects reported by the included studies. The quality of the evidence was characterized as high, moderate, low, or very low.” [Line 93-98]

We accordingly added the following sentence to the section of Results: “The quality of the evidence from the outcomes evaluated by the GRADE system was assessed as moderate (Supplementary Table S3). Substantial heterogeneity was the main reason responsible for the limited quality of the evidence.” [Line 132-134]

---

## [Decision Letter · Decision Letter 1]

28 Jun 2022

PONE-D-22-03407R1Effect of physical activity on the risk of frailty: a systematic review and meta-analysisPLOS ONE

Dear Dr. Liu,

Thank you for submitting your manuscript to PLOS ONE. After careful consideration, we feel that it has merit but does not fully meet PLOS ONE’s publication criteria as it currently stands. Therefore, we invite you to submit a revised version of the manuscript that addresses the points raised during the review process.

We look forward to receiving your revised manuscript.

Kind regards,

Mohammad Meshbahur Rahman, MS.

Academic Editor

PLOS ONE

Reviewers' comments:

Reviewer's Responses to Questions

**Comments to the Author**

1. If the authors have adequately addressed your comments raised in a previous round of review and you feel that this manuscript is now acceptable for publication, you may indicate that here to bypass the “Comments to the Author” section, enter your conflict of interest statement in the “Confidential to Editor” section, and submit your "Accept" recommendation.

Reviewer #1: All comments have been addressed

Reviewer #2: (No Response)

2. Is the manuscript technically sound, and do the data support the conclusions?

Reviewer #1: Yes

Reviewer #2: Partly

3. Has the statistical analysis been performed appropriately and rigorously? 

Reviewer #1: Yes

Reviewer #2: I Don't Know

4. Have the authors made all data underlying the findings in their manuscript fully available?

Reviewer #1: No

Reviewer #2: Yes

5. Is the manuscript presented in an intelligible fashion and written in standard English?

Reviewer #1: Yes

Reviewer #2: No

6. Review Comments to the Author

Reviewer #1: The manuscript is technically sound, statistically analysis was carefully conducted, nut i have not seen evidence that the authors have shared their data. The standard of English is improved

Reviewer #2: General comments

Thank you for addressing the reviewers’ comments in the updated version of the manuscript. Your study is interesting, and I believe it is important considering the impact of frailty on public health. I still find this study challenging to understand as the text is too long with important messages difficult to identify (e.g. certainty of evidence). Proofreading could assist, with focus on results and discussion.

Some specific comments

PRISMA checklist – Registration and protocol - Some of the items you classified as “NA - Not applicable”. This may not be correct as these items are applicable. However, I have the impression you failed to complete them. Please review this and be clear and say “not performed” for items 24 a,b,c if not performed by your team. Or, please explain why they are not applicable.

GRADE - I noticed you include GRADE as recommended – thank you. Yet, I missed seeing in your method/appendix a clear description of the GRADE parameters you used. Also, I believe the certainty of the evidence was not properly incorporated into your discussion, conclusion and abstract. Please review how GRADE is described, applied and reported across the manuscript to assist others in understanding the certainty of your findings.

Methods

Page 6, line 98: Ethical statement is likely irrelevant for this study and could be deleted.

Page 6, line 104: Please include a reference to assist readers in understanding how you are expressing your results. This is the first time I have come across “overall pooled effect estimates (ES)”. combining OR with HR. I have the impression this is not commonly issued nor recommended in the COCHRANE handbook. I could be wrong. Further details on your method may be required to guide readers.

Page 6, line 110. Please write in full “RR” as this is a new abbreviation

Results

I am concerned you may have included in your Meta-Analysis the same participants more than once (e.g. one participant may have been included in two analyses for different outcomes). For example, Yuki (2) and Yuki (3). Are these participants from the same Taiwan urban cohort with data collected in 2003? Are they the same participants? Please review this to make sure you do not include the same sample more than once in your pooling. Same for the Finish cohort.

Also, the comparison information in Figure S2 is challenging to understand. For example, you used the same name for comparison groups in different subgroups. You included “active vs inactive” for PA duration with no reference to duration. This seems unclear to me. Could you please rename them or provide further details in the figure.

Discussion/Conclusion

Please add a sentence about the certainty of evidence as this should be part of your main findings.

I hope some of these comments are helpful.

7. PLOS authors have the option to publish the peer review history of their article (what does this mean?). If published, this will include your full peer review and any attached files.

Reviewer #1: **Yes: **Dr. Reginald Quansah

Reviewer #2: No

---

## [Author Response · Author response to Decision Letter 1]

13 Jul 2022

We thank the reviewers and editors for his/her valuable comments, our point-by-point responses are shown below the comments. The changes made in the revised manuscript were highlighted in red.

C = comment; R = reply

#To reviewer 1

C1: The manuscript is technically sound, statistically analysis was carefully conducted, nut i have not seen evidence that the authors have shared their data. The standard of English is improved.

R1: Thanks for the reviewer’s comment. This study is a systematic review and meta-analysis; the data was extracted from published research. The data is available by contacting the corresponding author or can be extracted from original published research.

#To reviewer 2

C1: General comments: Thank you for addressing the reviewers’ comments in the updated version of the manuscript. Your study is interesting, and I believe it is important considering the impact of frailty on public health. I still find this study challenging to understand as the text is too long with important messages difficult to identify (e.g. certainty of evidence). Proofreading could assist, with focus on results and discussion.

R1: Thanks for reviewer’s comment. The whole manuscript has carefully revised accordingly. 

C2: Some specific comments: PRISMA checklist – Registration and protocol - Some of the items you classified as “NA - Not applicable”. This may not be correct as these items are applicable. However, I have the impression you failed to complete them. Please review this and be clear and say “not performed” for items 24 a,b,c if not performed by your team. Or, please explain why they are not applicable.

R2: Thanks for your suggestion, we have changed the “NA” with “not performed” in the PRISMA checklist.

C3: GRADE - I noticed you include GRADE as recommended – thank you. Yet, I missed seeing in your method/appendix a clear description of the GRADE parameters you used. Also, I believe the certainty of the evidence was not properly incorporated into your discussion, conclusion and abstract. Please review how GRADE is described, applied and reported across the manuscript to assist others in understanding the certainty of your findings.

R3: Thanks for your useful comment. We added detailed footnotes of the description of the GRADE approach for Table S3. Furthermore, we added the description of the certainty of the evidence across the manuscript accordingly.

In the abstract, we added the following sentences: 

 (1) “The quality of evidence was evaluated by using the Grading of Recommendations Assessment, Development and Evaluation (GRADE) approach.” [lines 9-11]

 (2) “and the GRADE approach classified the quality of evidence as moderate.” [line 12-13]

 (3) “This study with moderate-certainty evidence shows that…” [line 21]

In the conclusion, we added the following sentences:

 (1) “Based on available evidence with moderate quality involving 34,943 participants, results from this systematic review and meta-analysis suggest that a higher level of PA was significantly associated with a lower risk of frailty.” [186-188]

 (2) “In conclusion, this systematic review and meta-analysis with moderate-certainty evidence suggests that...” [line 274]

C4: Methods: Page 6, line 98: Ethical statement is likely irrelevant for this study and could be deleted.

R4: Thanks for your useful comment. We have deleted ethical statement in the manuscript.

C5: Page 6, line 104: Please include a reference to assist readers in understanding how you are expressing your results. This is the first time I have come across “overall pooled effect estimates (ES)”. combining OR with HR. I have the impression this is not commonly issued nor recommended in the COCHRANE handbook. I could be wrong. Further details on your method may be required to guide readers.

R5: Thanks for your important comment. We have included a reference to assist readers in understanding the methods [Ref. 36]. In fact, A Jatho et al. also conducted a meta-analysis by pooling the effect estimates using the adjusted OR, RR or HR and its 95% CI from each included study. Meanwhile, we have further conducted subgroup analysis by effect estimates (odds ratio, hazard ratio) to reveal the effect of the studies reporting OR and HR as a result, respectively.

C6: Page 6, line 110. Please write in full “RR” as this is a new abbreviation

R6: Thanks for your comment. We have changed “RR” with “ES” in the manuscript.

C7: Results: I am concerned you may have included in your Meta-Analysis the same participants more than once (e.g. one participant may have been included in two analyses for different outcomes). For example, Yuki (2) and Yuki (3). Are these participants from the same Taiwan urban cohort with data collected in 2003? Are they the same participants? Please review this to make sure you do not include the same sample more than once in your pooling. Same for the Finish cohort.

R7: Thanks for your comments. In fact, Yu et al. conducted the study from different areas: Yu (1) from Hong Kong, Yu (2) from Taiwan urban, and Yu (3) from Taiwan rural; hence, they are not the same population. The study from Yuk displayed the results from different physical activity indicators, we pooled the results and further conducted the subgroup analysis by physical activity indicators.

C8: Also, the comparison information in Figure S2 is challenging to understand. For example, you used the same name for comparison groups in different subgroups. You included “active vs inactive” for PA duration with no reference to duration. This seems unclear to me. Could you please rename them or provide further details in the figure.

R8: Thanks for your comments. In fact, the names of the comparison information were from the statement of the included studies. To help readers understand, we have renamed the comparison information concretely in Figure S2 and in Figure 2 accordingly. 

C9: Discussion/Conclusion: Please add a sentence about the certainty of evidence as this should be part of your main findings.

R9: Thanks for your suggestion. We added the description of the certainty of the evidence across the manuscript accordingly.

 We accordingly added the following sentence to the section of Discussion: ① “Based on available evidence with moderate quality involving 34,943 participants, results from this systematic review and meta-analysis suggest that a higher level of PA was significantly associated with a lower risk of frailty.” [Line 191-192]； ② “In conclusion, this systematic review and meta-analysis with moderate-certainty evidence suggests that….” [Line 279-282].

---

## [Decision Letter · Decision Letter 2]

18 Aug 2022

PONE-D-22-03407R2Effect of physical activity on the risk of frailty: a systematic review and meta-analysisPLOS ONE

Dear Dr. Liu,

Thank you for submitting your manuscript to PLOS ONE. After careful consideration, we feel that it has merit but does not fully meet PLOS ONE’s publication criteria as it currently stands. Therefore, we invite you to submit a revised version of the manuscript that addresses the points raised during the review process.

We look forward to receiving your revised manuscript.

Kind regards,

Mohammad Meshbahur Rahman, MS.

Academic Editor

PLOS ONE

Journal Requirements:

Reviewers' comments:

Reviewer's Responses to Questions

**Comments to the Author**

1. If the authors have adequately addressed your comments raised in a previous round of review and you feel that this manuscript is now acceptable for publication, you may indicate that here to bypass the “Comments to the Author” section, enter your conflict of interest statement in the “Confidential to Editor” section, and submit your "Accept" recommendation.

Reviewer #2: All comments have been addressed

2. Is the manuscript technically sound, and do the data support the conclusions?

Reviewer #2: Yes

3. Has the statistical analysis been performed appropriately and rigorously? 

Reviewer #2: I Don't Know

4. Have the authors made all data underlying the findings in their manuscript fully available?

Reviewer #2: Yes

5. Is the manuscript presented in an intelligible fashion and written in standard English?

Reviewer #2: Yes

6. Review Comments to the Author

Reviewer #2: The manuscript is well-written and presents relevant information for public health. It has improved substantially over the last rounds of reviews. I thank the authors for their patience and willingness to adjust the manuscript. Overall, the manuscript is excellent but I have only two small comments related to GRADE and Metanalyses.

GRADE approach for Table S3:

Please include all parameters with the description you used for GRADE (e.g. Phase of investigation, Study limitations, Inconsistency, Indirectness, Imprecision, Publication bias, Effect size).

Effect size: Could you please explain why you upgraded the evidence by 1x point due to the large effect? “2 Large effect: More than 2 studies showed the effect estimates<0.5 after adjusting for confounders. (marked 1 point)”. This parameter often considers the effect size of the pooling you are evaluating, not the estimates from individual studies. In this case, I believe the ES for the pooling is 0.59, which is above 0.5, suggesting no upgrade. Please review it.

Confounder: Is this related to imprecision? Please clarify this GRADE parameter as this is not clear to the readers. “ 3 Plausible confounding would change the effect: Reduced effect for ES<<1. (marked 2 points)”

Combining OR/HR/RR:

I believe OR/HR/RR should not be combined in one metanalysis without appropriate conversations. As this method has been selected for the main analysis, I recommended the main analysis be checked by an experienced statistician before publication. My impression is that the authors do not present enough evidence to support this choice/method. Combining OR/HR/RR is not the standard approach for metanalysis, and the evidence presented by the authors is insufficient (in my view). The reference Jatho et al used another reference that explains how random-effect metanalysis is performed. This paper used an example combining only ORs.

In sum, I am concerned with the method used but less concerned with the estimate presented. The combined estimates (OR/HR) are likely to be similar to the OR and HR estimates shown separately. Your review has several included papers reporting OR and HR. I don’t think it is necessary to present an unconventional metanalysis combining them because you have separate metanalysis for OR and HR showing similar estimates.

I hope some of these comments are helpful.

7. PLOS authors have the option to publish the peer review history of their article (what does this mean?). If published, this will include your full peer review and any attached files.

Reviewer #2: No

---

## [Author Response · Author response to Decision Letter 2]

7 Sep 2022

Re: Effect of physical activity on the risk of frailty: a systematic review and meta-analysis

We thank the reviewers and editors for his/her valuable comments, our point-by-point responses are shown below the comments. The changes made in the revised manuscript were highlighted in red.

C = comment; R = reply

# To Reviewer 2:

GRADE approach for Table S3: 

 C1: Please include all parameters with the description you used for GRADE (e.g. Phase of investigation, Study limitations, Inconsistency, Indirectness, Imprecision, Publication bias, Effect size). 

 R1: Thanks for your comment. We have included all the parameters that the GRADE handbook pointed out and added the footnotes for Table S3.

 C2: Effect size: Could you please explain why you upgraded the evidence by 1x point due to the large effect? “2 Large effect: More than 2 studies showed the effect estimates<0.5 after adjusting for confounders. (marked 1 point)”. This parameter often considers the effect size of the pooling you are evaluating, not the estimates from individual studies. In this case, I believe the ES for the pooling is 0.59, which is above 0.5, suggesting no upgrade. Please review it.

 R2: Thanks for your important comment. In fact, we have upgraded the evidence by 1 point according the GRADE handbook, which pointed out that the effect <0.5 based on direct evidence could be considered as large effect.

 C3: Confounder: Is this related to imprecision? Please clarify this GRADE parameter as this is not clear to the readers. “3 Plausible confounding would change the effect: Reduced effect for ES<<1. (marked 2 points)”

 R3: Thanks for your important comment. In fact, this point was pointed out according to Factors that can increase the quality of the evidence in the GRADE handbook. Plausible confounding that would reduce effect for ES<<1 could upgrade by one level. We updated the Table S3 and furtherly updated the footnotes.

 C4：Combining OR/HR/RR: I believe OR/HR/RR should not be combined in one metanalysis without appropriate conversations. As this method has been selected for the main analysis, I recommended the main analysis be checked by an experienced statistician before publication. My impression is that the authors do not present enough evidence to support this choice/method. Combining OR/HR/RR is not the standard approach for metanalysis, and the evidence presented by the authors is insufficient (in my view). The reference Jatho et al used another reference that explains how random-effect metanalysis is performed. This paper used an example combining only ORs. In sum, I am concerned with the method used but less concerned with the estimate presented. The combined estimates (OR/HR) are likely to be similar to the OR and HR estimates shown separately. Your review has several included papers reporting OR and HR. I don’t think it is necessary to present an unconventional metanalysis combining them because you have separate metanalysis for OR and HR showing similar estimates.

 R4: Thanks for your helpful comment. Indeed, there is limited evidence combined OR/HR/RR in a mate-analysis. We have consulted experienced statisticians at Sun Yat-sen University. Considering that our study included 12 records using OR as effect estimates and only 2 records using HR as effect estimates, we pooled the effect estimates using the adjusted OR/HR and its 95% CI from each included study as Jatho et al did in the main analysis. In the meanwhile, to differ the effect of the studies reporting OR and HR, we further conducted subgroup analysis by effect estimates. For the results of subgroup analysis did not changed remarkably, we still used the results of the main analysis with relatively more included studies.

---

## [Decision Letter · Decision Letter 3]

5 Oct 2022

PONE-D-22-03407R3Effect of physical activity on the risk of frailty: a systematic review and meta-analysisPLOS ONE

Dear Dr. Liu,

Thank you for submitting your manuscript to PLOS ONE. After careful consideration, we feel that it has merit but does not fully meet PLOS ONE’s publication criteria as it currently stands. Therefore, we invite you to submit a revised version of the manuscript that addresses the points raised during the review process.

We look forward to receiving your revised manuscript.

Kind regards,

Mohammad Meshbahur Rahman, MS.

Academic Editor

PLOS ONE

Reviewers' comments:

Reviewer's Responses to Questions

**Comments to the Author**

1. If the authors have adequately addressed your comments raised in a previous round of review and you feel that this manuscript is now acceptable for publication, you may indicate that here to bypass the “Comments to the Author” section, enter your conflict of interest statement in the “Confidential to Editor” section, and submit your "Accept" recommendation.

Reviewer #2: (No Response)

Reviewer #3: (No Response)

2. Is the manuscript technically sound, and do the data support the conclusions?

Reviewer #2: Partly

Reviewer #3: Yes

3. Has the statistical analysis been performed appropriately and rigorously? 

Reviewer #2: No

Reviewer #3: Yes

4. Have the authors made all data underlying the findings in their manuscript fully available?

Reviewer #2: Yes

Reviewer #3: Yes

5. Is the manuscript presented in an intelligible fashion and written in standard English?

Reviewer #2: Yes

Reviewer #3: Yes

6. Review Comments to the Author

Reviewer #2: Thank you for attempting to address my questions and concerns in your response. Below I list some of my remaining concerns as I believe the information provided in your response was insufficient and not well supported by the literature. I imagine it may be time-consuming to address these comments. Apologies for this fourth revision. Yet, I cannot recommend this paper for publication if I have concerns with critical aspects of the method and main findings. I would appreciate it if you could review the points below:

C1: GRADE description

Page 6, line 97 – I still find it difficult to understand your GRADE findings. In the method, you mentioned that GRADE was used but did not provide a description of each parameter. Again, in table S3, you mention the GRADE decisions with no descriptions of the parameters (e.g. Study limitations: No serious risk of bias was found. Indirectness: no. Imprecision: no.). Based on the current information provided in the manuscript, it is not possible for readers to understand how you applied GRADE in your study. This needs to be corrected before publication

Please provide a clear description for each GRADE parameter in your method or Table S3 (e.g. Imprecision: We downgraded if there were <400 participants).

C2: Effect size upgrade (related to what I described above).

It is still unclear if you have upgraded (or not) due to the large magnitude of the effect. Could you please review this? I have the impression you should not upgrade this parameter. Your estimate of effect is 0.59 (0.51 to 0.67), which is above 0.5 (the suggested parameter to upgrade is <0.5). Therefore, it should not be upgraded based on effect size. Please note that none of your estimates presents in Table 2 is below 0.5.

C4: Combining OR/HR. There is no quality evidence supporting your method to combine OR/HR in your main metanalysis. You also mentioned that the evidence supporting this approach is limited. If there is no high-quality evidence supporting this decision, please only present the meta-analysis for OR and HR separately. Please note the main findings of your review is based on a questionable Meta-Analysis, and, in my view, this is concerning. I am happy to accept this method if the Editor supports your decision.

Abstract findings: Most of your results are ORs and the interpretation should be framed in terms of odds, not in terms of risk. ORs help to understand the magnitude of an effect but are often mistaken for relative risk ratios. Please review your abstract to express your results as odds. Some references:

Norton EC, Dowd BE, Maciejewski ML. Odds Ratios—Current Best Practice and Use. JAMA. 2018;320(1):84–85. doi:10.1001/jama.2018.6971

Andrade C. Understanding relative risk, odds ratio, and related terms: as simple as it can get. J Clin Psychiatry. 2015 Jul;76(7):e857-61. doi: 10.4088/JCP.15f10150. PMID: 26231012.

Once again, I hope some of these comments/suggestions are helpful.

Reviewer #3: The research procedure of this paper is standardized and the process is comprehensive, which is a complete meta-analysis relatively. However, I note that the authors searched only three databases during the literature search, including PubMed, Embase, and Web of science, omitting The Cochrane Library database. In addition, several important Chinese databases have been omitted, such as CNKI, WanFang Data and the VIP database, and the included studies included samples from Hong Kong and Taiwan, China. How did the authors ensure the scientific validity of the findings?

7. PLOS authors have the option to publish the peer review history of their article (what does this mean?). If published, this will include your full peer review and any attached files.

Reviewer #2: No

Reviewer #3: No

---

## [Author Response · Author response to Decision Letter 3]

13 Oct 2022

Re: Effect of physical activity on the risk of frailty: a systematic review and meta-analysis (PONE-D-22-03407R3)

C = Comment， R = Reply

To Reviewer #2: 

C1: GRADE description: Page 6, line 97 – I still find it difficult to understand your GRADE findings. In the method, you mentioned that GRADE was used but did not provide a description of each parameter. Again, in table S3, you mention the GRADE decisions with no descriptions of the parameters (e.g. Study limitations: No serious risk of bias was found. Indirectness: no. Imprecision: no.). Based on the current information provided in the manuscript, it is not possible for readers to understand how you applied GRADE in your study. This needs to be corrected before publication. Please provide a clear description for each GRADE parameter in your method or Table S3 (e.g. Imprecision: We downgraded if there were <400 participants).

R1: Thanks for reviewer’s suggestion. We have modified the Table S3, so the information can be clearer displayed. 

We also added the following description in the results section: “By using GRADE approach, it is found that substantial heterogeneity was the main reason responsible for the limited quality of the evidence, whereas all plausible confounding factors were considered (Supplementary Table S3). Hence, the quality of the evidence from the outcomes evaluated by the GRADE system was assessed as low as a whole.” [Lines: 134-138]

C2: Effect size upgrade (related to what I described above). It is still unclear if you have upgraded (or not) due to the large magnitude of the effect. Could you please review this? I have the impression you should not upgrade this parameter. Your estimate of effect is 0.59 (0.51 to 0.67), which is above 0.5 (the suggested parameter to upgrade is <0.5). Therefore, it should not be upgraded based on effect size. Please note that none of your estimates presents in Table 2 is below 0.5.

R2. Thanks for reviewer’s suggestion. We have modified the Table S3, so the information can be clearer displayed. We also added the following description in the results section: “By using GRADE approach, it is found that substantial heterogeneity was the main reason responsible for the limited quality of the evidence, whereas all plausible confounding factors were considered (Supplementary Table S3). Hence, the quality of the evidence from the outcomes evaluated by the GRADE system was assessed as low as a whole.” [Lines: 134-138]

C3: Combining OR/HR. There is no quality evidence supporting your method to combine OR/HR in your main metanalysis. You also mentioned that the evidence supporting this approach is limited. If there is no high-quality evidence supporting this decision, please only present the meta-analysis for OR and HR separately. Please note the main findings of your review is based on a questionable Meta-Analysis, and, in my view, this is concerning. I am happy to accept this method if the Editor supports your decision.

R3: Thanks for reviewer’s comments. In our study we only included cohort studies, though two methods of logistic regression and Cox regression were used among these studies. We calculated the whole effects by synthesizing results from all included studies (ES=0.59, 95% CI= 0.51-0.67), and did the stratified analysis according their analyzed methods (for logistic regression: ES=0.60, 95% CI= 0.52-0.70); for cox regression, ES =0.54, 95% CI= 0.38-0.77). We believe this can enhance the robustness and credibility of the results.

C4: Abstract findings: Most of your results are ORs and the interpretation should be framed in terms of odds, not in terms of risk. ORs help to understand the magnitude of an effect but are often mistaken for relative risk ratios. Please review your abstract to express your results as odds. Some references: Norton EC, Dowd BE, Maciejewski ML. Odds Ratios—Current Best Practice and Use. JAMA. 2018;320(1):84–85. doi:10.1001/jama.2018.6971. Andrade C. Understanding relative risk, odds ratio, and related terms: as simple as it can get. J Clin Psychiatry. 2015 Jul;76(7):e857-61. doi: 10.4088/JCP.15f10150. PMID: 26231012. Once again, I hope some of these comments/suggestions are helpful.

R4. Thanks for reviewer’s helpful suggestion. We have modified the abstract accordingly. The “risk” was replaced with the “odds”. We also update the description in the sections of results and discussion.

To Reviewer #3:

C1: The research procedure of this paper is standardized and the process is comprehensive, which is a complete meta-analysis relatively. However, I note that the authors searched only three databases during the literature search, including PubMed, Embase, and Web of science, omitting The Cochrane Library database. In addition, several important Chinese databases have been omitted, such as CNKI, WanFang Data and the VIP database, and the included studies included samples from Hong Kong and Taiwan, China. How did the authors ensure the scientific validity of the findings?

R1: Thanks for reviewer’s comments. We only considered PubMed, Embase, and Web of science, this was because that almost the vast majority of high-quality Chinese Journals are included into these three datasets. The Cochrane Library database was also considered, as PubMed also included the Cochrane Database of Systematic Reviews. Hence, searched three databases of PubMed, Embase, and Web of science could include nearly all publications. In addition, a reverse reference citation tracking was also carried out to search for the possible literature.

---

## [Decision Letter · Decision Letter 4]

14 Nov 2022

Effect of physical activity on the risk of frailty: a systematic review and meta-analysis

PONE-D-22-03407R4

Dear Dr. Liu,

We’re pleased to inform you that your manuscript has been judged scientifically suitable for publication and will be formally accepted for publication once it meets all outstanding technical requirements.

Kind regards,

Mohammad Meshbahur Rahman, MS.

Academic Editor

PLOS ONE

Additional Editor Comments (optional):

I want to thank all authors for extensively revising the manuscript. The current version of manuscript found impressive and publishable in PLOS ONE.

Reviewers' comments:

Reviewer's Responses to Questions

**Comments to the Author**

1. If the authors have adequately addressed your comments raised in a previous round of review and you feel that this manuscript is now acceptable for publication, you may indicate that here to bypass the “Comments to the Author” section, enter your conflict of interest statement in the “Confidential to Editor” section, and submit your "Accept" recommendation.

Reviewer #3: (No Response)

2. Is the manuscript technically sound, and do the data support the conclusions?

Reviewer #3: (No Response)

3. Has the statistical analysis been performed appropriately and rigorously? 

Reviewer #3: (No Response)

4. Have the authors made all data underlying the findings in their manuscript fully available?

Reviewer #3: (No Response)

5. Is the manuscript presented in an intelligible fashion and written in standard English?

Reviewer #3: (No Response)

6. Review Comments to the Author

Reviewer #3: (No Response)

7. PLOS authors have the option to publish the peer review history of their article (what does this mean?). If published, this will include your full peer review and any attached files.

Reviewer #3: No

---

## [Editor Report · Acceptance letter]

21 Nov 2022

PONE-D-22-03407R4 

Effect of physical activity on the risk of frailty: a systematic review and meta-analysis 

Dear Dr. Liu:

I'm pleased to inform you that your manuscript has been deemed suitable for publication in PLOS ONE. Congratulations! Your manuscript is now with our production department. 

Kind regards, 

on behalf of

Mr. Mohammad Meshbahur Rahman 

Academic Editor

PLOS ONE